# Prevalence of biofilms in *Candida* spp. bloodstream infections: A meta-analysis

**María Belén Atiencia-Carrera**[1]☯, **Fausto Sebastián Cabezas-Mera**[1]☯, **Eduardo Tejera**[2]\*, **António Machado**[1]\*

**1** Instituto de Microbiología, Colegio de Ciencias Biológicas y Ambientales (COCIBA), Universidad San Francisco de Quito (USFQ), Diego de Robles y Vía Interoceánica, Campus Cumbayá, Quito, Pichincha, Ecuador, **2** Facultad de Ingeniería y Ciencias Agropecuarias Aplicadas, Grupo de Bioquimioinformática, Universidad de Las Américas, Quito, Pichincha, Ecuador

☯ These authors contributed equally to this work.

\* eduardo.tejera@udla.edu.ec (ET); amachado@usfq.edu.ec (AM)

## Abstract

### Context

*Candida*-related infections are nowadays a serious Public Health Problem emerging multi-drug-resistant strains. *Candida* biofilm also leads bloodstream infections to invasive systemic infections.

### Objective

The present meta-analysis aimed to analyze *Candida* biofilm rate, type, and antifungal resistance among hospitalized patients between 1995 and 2020.

### Data sources

Web of Science, Scopus, PubMed, and Google Scholar databases were searched for English papers using the following medical subject heading terms (MESH): "invasive candidiasis"; "bloodstream infections"; "biofilm formation"; "biofilm-related infections"; "mortality"; and "prevalence".

### Study selection

The major inclusion criteria included reporting the rate of biofilm formation and the prevalence of biofilm-related to *Candida* species, including observational studies (more exactly, cohort, retrospective, and case-control studies). Furthermore, data regarding the mortality rate, the geographical location of the study set, and the use of anti-fungal agents in clinical isolates were also extracted from the studies.

### Data extraction

Independent extraction of articles by 2 authors using predefined data fields, including study quality indicators.

**Data Availability Statement:** All relevant data are within the paper and its Supporting Information files.

**Funding:** This work was supported by COCIBA Research Grant 2018-2019 through project ID:

12260 entitled "Adhesión inicial y resistencia antimicrobiana de Candida sp. aisladas de la microbiota humana", under regulations of the Ministry of Health of Ecuador (Contrato Marco de Acceso a los Recursos Genéticos No. MAE-DNB-CM-2016-0046).

**Competing interests:** The authors have declared that no competing interests exist.

## Data synthesis

A total of 31 studies from publicly available databases met our inclusion criteria. The biofilm formation in the data set varied greatly from 16 to 100% in blood samples. Most of the studies belonged to Europe (17/31) and Asia (9/31). Forest plot showed a pooled rate of biofilm formation of 80.0% (CI: 67–90), with high heterogeneity (Q = 2567.45, $I^2$ = 98.83, $\tau^2$ = 0.150) in random effects model ($p$ < 0.001). The funnel plot and Egger's linear regression test failed to find publication bias ($p$ = 0.896). The mortality rate in *Candida*-related bloodstream infections was 37.9% of which 70.0% were from biofilm-associated infections. Furthermore, *Candida* isolates were also characterized in low, intermediate, or high biofilm formers through their level of biofilm mass (crystal violet staining or XTT assays) after a 24h growth. When comparing between countries, statistical differences were obtained ($p$ = 0.0074), showing the lower and higher biofilm prevalence values in Italy and Spain, respectively. The prevalence of low, intermediate, and high biofilms were 36.2, 18.9, and 35.0% ($p$ < 0.0001), respectively. *C. tropicalis* was the prevalent species in high biofilm formation (67.5%) showing statistically significant differences when compared to other *Candida* species, except for *C. krusei* and *C. glabrata*. Finally, the rates of antifungal resistance to fluconazole, voriconazole, and caspofungin related to biofilm were 70.5, 67.9 and 72.8% ($p$ < 0.001), respectively.

## Conclusions

Early detection of biofilms and a better characterization of *Candida* spp. bloodstream infections should be considered, which eventually will help preserve public health resources and ultimately diminish mortality among patients.

## Introduction

Invasive candidiasis is a systemic mycosis caused by *Candida* species, being commonly described as an opportunistic infection. The population group more vulnerable for invasive candidiasis includes patients with critical illness or immunosuppression (such as hematological and solid organ malignancy, hematopoietic cell and solid organ transplantation, recent abdominal surgery, and hemodialysis), or even people with a central venous catheter, parenteral nutrition. In addition, people that received broad-spectrum antibiotics or with drug habits are also susceptible to invasive candidiasis, as well as premature newborns [1]. All these plausible scenarios lead this systemic infection to be nowadays the 4th leading nosocomial infection in the United States, demonstrating mortality of up to 40% [2]. In Europe, Bassetti and colleagues realized a multinational and multicenter study in 2019 reporting 7.07 episodes per 1000 in European intensive care units (ICUs) with a 30-day mortality of 42% [3]. While, in the Asia-Pacific region, Hsueh and colleagues reported a candidemia incidence in ICUs of 5- to 10-fold higher than in the entire hospital and a mortality rate of patients between 35% and 60% [4]. In Latin America, Nucci and colleagues realized a laboratory-based survey between November 2008 and October 2010 among 20 tertiary care hospitals in seven Latin American countries, reporting an overall incidence of 1.18 cases per 1,000 in general admissions [5]. The mortality associated with invasive candidiasis is similar or even higher in other worldwide countries [6].

To understand the dimension of this infection and its virulence, we must define the term invasive candidiasis as both forms of candidemia detected in the blood and tissues or deep organs under the mucosal surfaces (also known as deep candidiasis). Deep candidiasis can remain localized or spread causing a secondary infection [7]. Patients with a systemic infection induced by *Candida* spp. can be subdivided into three groups: (1) those who present with bloodstream infection (candidemia); (2) those who develop deep-seated candidiasis (most frequently intra-abdominal candidiasis); and, (3) those who develop a combination of these two groups [8].

The gold standard for the diagnosis of invasive candidiasis is the growth culture, being blood culture commonly used to diagnose candidemia while culture media is applied to diagnose deep candidiasis from tissue biopsies [9]. In this meta-analysis, we only evaluated studies using positive blood cultures to evaluate the biofilm formation and other related factors in candidiasis virulence. More exactly, the selected studies performed an *in vitro* biofilm assay using *Candida* isolates from blood samples of the patients with catheter-related candidemia (CRC) and non-CRC. In cases of patients with CRC, the standard procedure was blood cultures from obtained the catheter and peripheral veins, whereas non-CRC was indicated by the recovery of *Candida* spp. from only blood samples, as previously described by Guembe and colleagues [10].

Nosocomial infections are closely associated with biofilms growing attached to medical devices or host tissues [11]. Biofilms are the predominant growth state of many microorganisms, being a community of irreversible adherent cells with different phenotypic and structural properties when compared to free-floating (planktonic) cells. National Institutes of Health estimated that biofilms are responsible, in one way or another, for more than 80% of all microbial infections in the United States [12]. *Candida* species can produce well-structured biofilms composed of multiple types of cell and even microbial species, leading to an intrinsic resistance against a wide variety of stress factors, such as various antifungal drugs and immune defense mechanisms [13]. Although the dynamics biofilm-host is not yet fully understood, it is well-known that *Candida* biofilms inhibit the innate immune system of the host [14]. Therefore, our main goal was to analyze the relationship between biofilms and mortality in *Candida* spp. related infections, showing a severe menace to the Public Health System with serious outcomes.

## Results

### Study inclusion criteria and characteristics of the eligible studies

A total of 214 studies were retrieved and 70 full texts were reviewed from publicly available databases (Web of Science, Scopus, PubMed, and Google Scholar). Thirty-one studies met our inclusion criteria (Fig 1). The final data set included studies covering different global regions (most of them in Europe). All available and relevant data were extracted of each study, more exactly, biofilm rate, biofilm type, underlying disease of the patients, *Candida* species reported, and antifungal resistance. The data was then used to create other databases, collecting information of at least five or more papers, and consequently, each paper was cited more than once. These additional databases were chosen to realize subgroup analysis using a random-effect model and to answer relevant questions about *Candida*-related biofilms, such as the mortality rate related to biofilms, the geographical distribution of biofilms, the characterization of biofilm production among *Candida* species, and the correlation between biofilm formation and antifungal resistance (S1 and S2 Files).

As shown in Fig 1, a total data set of 31 studies was achieved for the present meta-analysis following the eligibility criteria, screening process, and quality assessment.

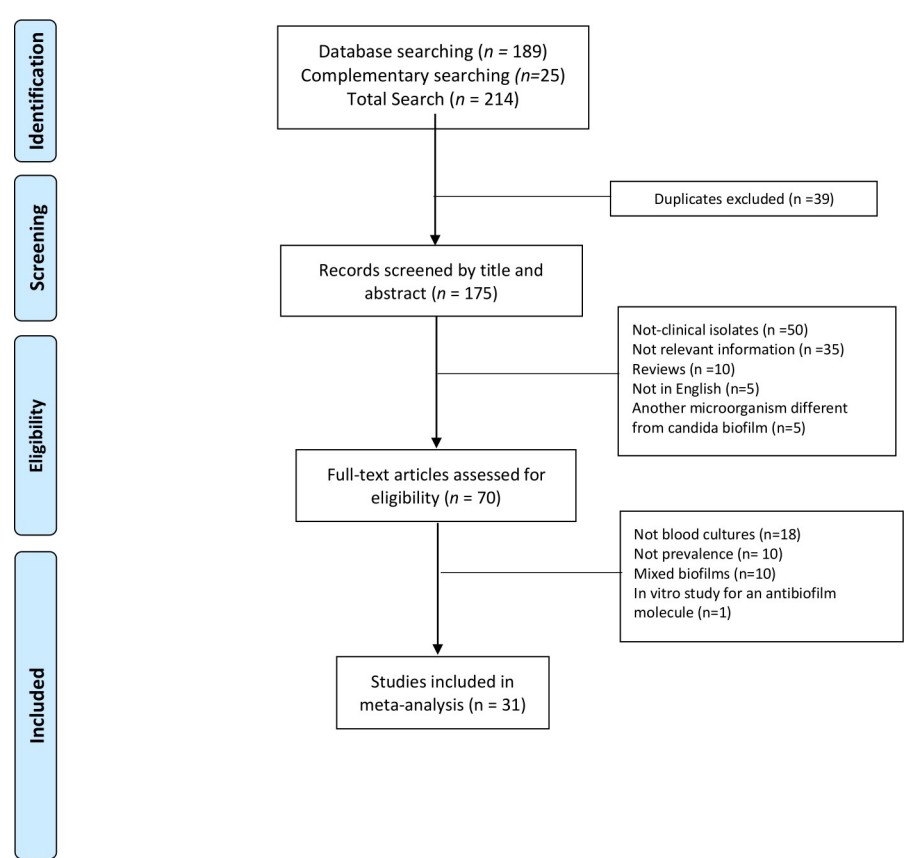

**Fig 1. Prisma flow chart of included and excluded studies of the selection process.**

## Overall effects of *Candida* biofilms

The data set reported biofilm rates of *Candida*-related infections among hospitalized patients between 1995 and 2020 in several countries worldwide. As shown in Table 1, the biofilm formation by *Candida* spp. isolates in the data set varied greatly from 16% to 100% in blood samples from hospitalized patients. Most of the data set belonged to studies realized in Europe (17/31), followed by Asia (9/31), South America (3/31), and North America (2/31).

Although the methodologies to quantify biofilm biomass varied between studies, these methodologies are based on the optical density (OD) obtained by the combination of a certain colorimetric compound or a simple dissolution in a buffer or water with the growth of the isolated *Candida* sp. and then it's compared with reference *Candida* strains in the same growth conditions. The main methodologies in our study set were crystal violet (CV) assays using microplate reader (51.6%; 16/31), assays with tetrazolium dye (2,3-bis-(2-methoxy-4-nitro-5-sulphenyl)-(2H)-tetrazolium-5-carboxanilide, XTT) using micro plate reader (35.5%; 11/31), and Branchini's method (9.7%; 3/31). The Branchini's method, also called slime production method, is based on the production of a viscid slime layer by the growth of the *Candida* isolate in a tube containing Sabouraud broth [15].

Regardless of the applied methodology in the studies, all these authors were able to evaluate biofilm formation among *Candida* isolates. However, only 18 of 31 studies were able to categorize the biofilm formation, and so just 5 studies were able to evaluate a positive correlation between biofilm presence and increment of antifungal resistance in the treatment. Finally, the

**Table 1. General information extracted from the data set selected for the present meta-analysis.**

| First author | Publication (year) | Region | Country | Methodology to measure biofilm | Biofilm rate, n (%) | Biofilm formation, n (%) | | | Correlation between biofilm and resistance | Attributable mortality, n (%) |
|---|---|---|---|---|---|---|---|---|---|---|
| | | | | | | High | Medium | Low | | |
| Atalay | 2015 | Asia | Turkey | CV (450 nm) | 8/50 (16) | | | | No | |
| Tumbarello | 2007 | Europe | Italy | PBS (405 nm) & XTT (490 nm) | 80/294 (27.2) | | | | No | 56 (70.0) |
| Tortonaro | 2013 | Europe | Italy | XTT (490 nm) | 160/451 (35.4) | 116 (72.5) | | 44 (27.5) | No | 11 (6.9) |
| Banerjee | 2015 | Asia | India | Branchini's method | 31/80 (38.8) | | | | No | 5 (16.1) |
| Tumbarello | 2012 | Europe | Italy | PBS (405 nm) & XTT (490 nm) | 84/207 (40.6) | | | | No | 43 (51.2) |
| Pongracz | 2016 | Europe | Hungary | CV (570 nm) & XTT (490 nm) | 43/93 (46.2) | 12 (27.9) | | 31 (72.1) | Yes | 23 (53.49) |
| Sida | 2015 | Asia | India | Branchini's method | 2/4 (50) | | | | No | |
| Rodrigues | 2019 | South America | Brazil | Christensen's method | 15/28 (53.8) | | | | No | 6 (40.0) |
| Gangneux | 2018 | Europe | France | BioFilm Ring Test | 181/319 (56.7) | 132 (72.9) | | 49 (27.1) | No | 55 (30.4) |
| Shin | 2002 | Asia | Korea | DW (405 nm) | 58/101 (57.4) | | | | No | |
| Pannanusorn | 2012 | Europe | Sweden | XTT (590 nm) | 231/393 (58.7) | 101 (43.7) | | 130 (56.3) | No | |
| Tascini | 2018 | Europe | Italy | XTT (490 nm) | 57/89 (64.0) | | | | No | 25 (43.9) |
| Tobudic | 2011 | Europe | Austria | CV (630 nm), PBS (405 nm) & XTT (620 nm) | 34/47 (72.3) | | | | No | 18 (52.9) |
| Tulasidas | 2018 | Asia | India | CV (570 nm) | 55/74 (74.3) | | | | No | |
| Pfaller | 1995 | North America | USA | Branchini's method | 13/17 (76.5) | 3 (23.1) | 6 (46.1) | 4 (30.8) | No | |
| Pham | 2019 | Asia | Thailand | XTT (490 nm) | 38/46 (76.4) | 25 (65.8) | | 13 (34.2) | No | 13 (34.2) |
| Guembe | 2014 | Europe | Spain | CV (550 nm) | 45/54 (76.4) | | | | No | |
| Kumar | 2006 | Asia | India | UPW (405 nm) | 30/36 (83.3) | | | | No | |
| Rajendran | 2016 | Europe | Scotland | CV (570 nm) | 245/280 (87.7) | 56 (22.9) | 44 (17.9) | 144 (58.9) | Yes | |
| Stojanovic | 2015 | Europe | Serbia | CV (595 nm) | 7/8 (87.5) | 2 (28.6) | 3 (42.8) | 2 (28.6) | Yes | |
| Turan | 2018 | Asia | Turkey | CV (540 nm) | 145/162 (89.5) | 37 (25.5) | 61 (42.1) | 47 (32.4) | Yes | |
| Tulyaprawat | 2020 | Asia | India | XTT (490 nm) | 45/48 (93.8) | 26 (57.8) | | 19 (42.2) | No | |
| Muñoz | 2018 | Europe | Spain | CV (540 nm) | 280/280 (100.0) | 90 (32.1) | 190 (67.9) | | No | 95 (33.9) |
| Soldini | 2017 | Europe | Italy | CV (540 nm) | 190/190 (100.0) | 84 (44.2) | 38 (20.0) | 68 (35.8) | No | 89 (46.8) |
| Vitális | 2020 | Europe | Hungary | CV (550 nm) | 127/127 (100.0) | 28 (22.0) | 69 (54.4) | 30 (23.6) | No | 70 (55.1) |
| Prigitano | 2013 | Europe | Italy | XTT (490 nm) | 297/297 (100.0) | 96 (32.3) | 141 (47.5) | 60 (20.2) | No | 65 (21.9) |

(*Continued*)

**Table 1.** (Continued)

| First author | Publication (year) | Region | Country | Methodology to measure biofilm | Biofilm rate, n (%) | Biofilm formation, n (%) | | | Correlation between biofilm and resistance | Attributable mortality, n (%) |
|---|---|---|---|---|---|---|---|---|---|---|
| | | | | | | High | Medium | Low | | |
| Treviño-Rangel | 2018 | North America | México | CV (595 nm) | 89/89 (100.0) | | | | No | 32 (35.9) |
| Marcos-Zambrano | 2017 | Europe | Spain | CV (540 nm) | 22/22 (100.0) | | 13 (59.1) | 9 (40.9) | Yes | 3 (13.6) |
| Marcos-Zambrano | 2014 | Europe | Spain | CV (540 nm) | 564/564 (100.0) | 194 (34.4) | 187 (33.1) | 181 (32.1) | No | |
| Thomaz | 2019 | South America | Brazil | CV (595 nm) & XTT (490 nm) | 38/38 (100.0) | 3 (7.9) | | 35 (92.1) | No | |
| Herek | 2019 | South America | Brazil | CV (570 nm) | 13/13 (100.0) | 3 (23.1) | 7 (53.8) | 3 (23.1) | No | |

The prevalence of biofilm formation was calculated with 95% CI through random-model and significance level ≤0.05 (*p*-value). The sample size and prevalence were used to calculate the combined biofilm produced. Attribute mortality was calculated by the number of deaths among patients with biofilm in blood samples. The information summarized in the table did not show information on the patients' underlying diseases and resistance. The methodologies used to measure biofilm in the studies were based in the optical density (nm, i.e., wavelength in the assay) of the biomass from growth culture, more exactly: XTT—using micro plate reader with yellow tetrazolium salt; CV—using micro plate reader with crystal violet staining; UPW—using micro plate reader with ultra-pure water; DW—using microplate reader with distilled water; Branchini's method—evaluating the adherent growth of the biofilm's slime production; BioFilm Ring Test—using micro plate reader with a BioFilm Index (BFI) software; and, Christensen's method—evaluating the adherent growth of the biofilm in Falcon tube with safranin or trypan blue staining.

incidence of mortality among patients varied considerably among studies, reporting the values of attributable mortality between 6.9 and 70%. All the information extracted is available in the supplementary section.

Analysis of the forest plot was then realized with data set, showing a pooled rate of biofilm formation of 80.0% (CI: 67–90), as shown in Fig 2. The heterogeneity indices obtained using random effects model ($p < 0.001$) were Q = 2567.45 ($p < 0.001$), $I^2 = 98.83$, and $\tau^2 = 0.150$.

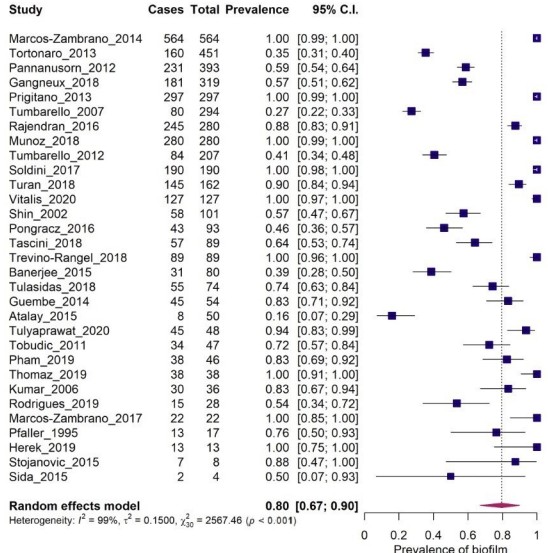

**Fig 2. Forest plot of the meta-analysis of the prevalence of biofilm formation in *Candida* spp. isolated from blood clinical samples.**

The pooled rate of biofilm formation obtained needs to be carefully analyzed given the high value of heterogeneity. This will be addressed in our discussion.

A funnel plot was realized to evaluate the existence of publication bias in the final data set (Fig 3). Furthermore, Egger's linear regression test was also used to reveal any publication bias and possible asymmetric data distribution in the funnel plot.

No publication bias was identified by the Egger's linear regression test ($p = 0.896$). However, as we will discuss in the next section the qualitative analysis of the funnel clearly suggests some biases from the departure of the geometry from the expected triangular form. The funnel plot of this study illustrates the effect size (biofilm prevalence) on the x-axis and the standard error (SE) on the y-axis. In case of no publication bias in the data set, the studies are distributed evenly around the pooled effect size. The smaller studies should appear near the bottom due to their higher variance when compared to the larger studies, which should be placed at the top of the plot. The diagonal lines show the expected 95% confidence intervals around the summary estimate. In the absence of heterogeneity, the studies of the data set should lie within the funnel defined by these diagonal lines. However, heterogeneity and some asymmetries among the studies of the data set were illustrated by the funnel plot. In our case, we found studies with low errors (similar sizes) but with drastic differences in the biofilm prevalence. This type of pattern probably indicates the presence of confounding variables (sub-groups undelaying structures) which are not included in the global analysis.

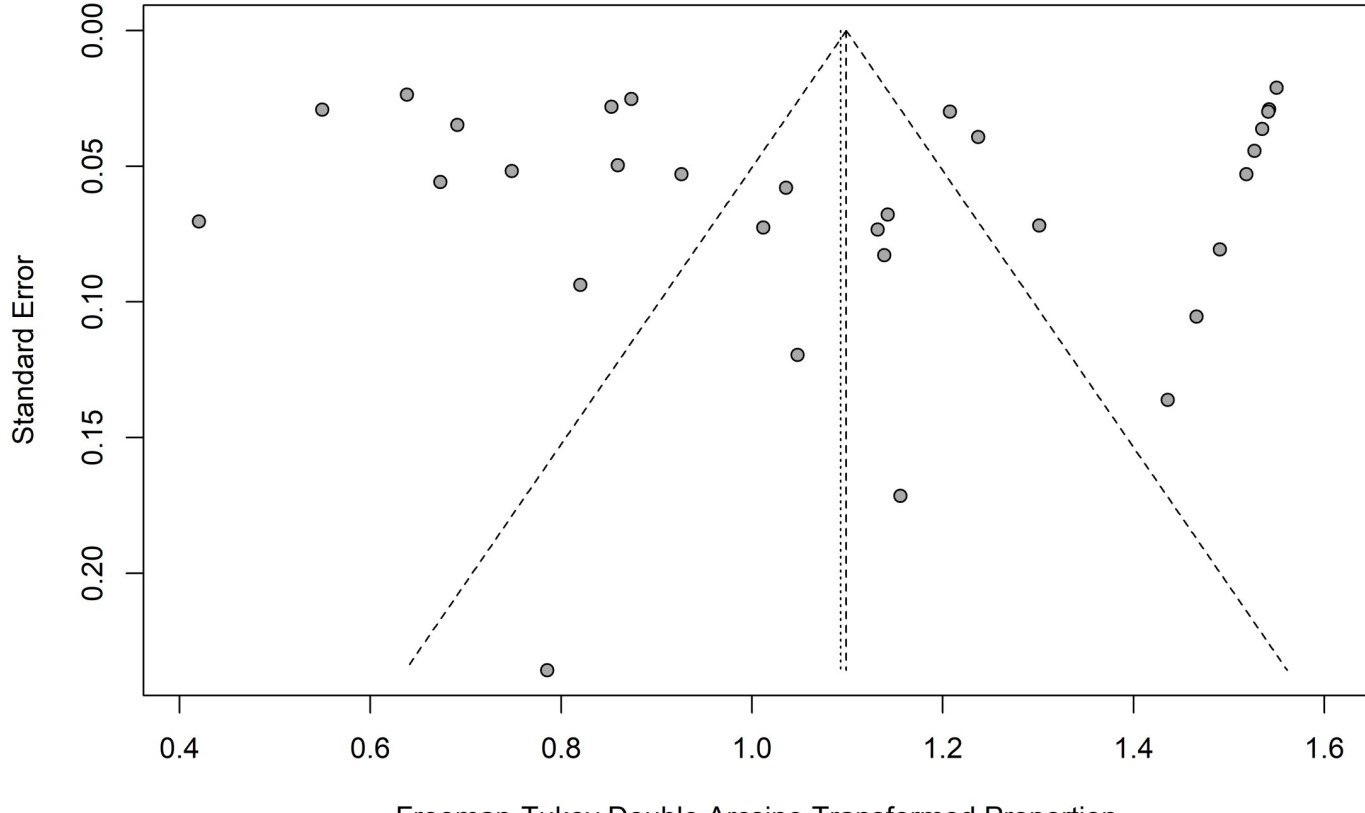

**Fig 3. Funnel plot of the meta-analysis on the biofilm formation rate in *Candida* spp. isolated from blood clinical samples.** Studies are represented by a point. The X-axis represents the effect size (biofilm prevalence), and the Y-axis shows the standard error. Despite some asymmetry revealed by the funnel plot in the data set, Egger's test failed to show publication bias ($p = 0.896$).

Although an obvious biofilm prevalence was found in the data set, the selected studies poorly described the underlying conditions of the patient with biofilm production. The analysis of these conditions among the patients was merely descriptive, as shown in Table 2.

The lack of a detail description of the clinical background and host factors in the patients among the studies represents a main drawback of the present meta-analysis precluding the evaluation of clinical or patient factors and the ability of *Candida* isolates to establish biofilm. Nonetheless, the ability to establish biofilm is a virulence factor by itself and should be evaluate as risk factor in the treatment of patients with *Candida*-related blood infections. As summarized in Table 2, only 16 of 31 studies reported some sort of clinical background of the patients with *Candida*-related bloodstream infections. From this subset of studies, patients evidenced mainly the following clinical conditions: hematological or solid cancer (68.8%, 11/16), surgery interventions (62.5%, 10/16); patients with central venous catheter (56.3%, 9/16); adults under total parenteral nutrition (50.0%, 8/16); patients with human immunodeficiency virus (HIV; 50.0%, and 8/16); patients with diabetes (43.8%; 7/16); patients in the intensive care unit (ICU; 37.5%, and 6/16); patients with immunosuppressive therapy (37.5%, 6/16) and, the remaining clinical backgrounds were only described in 25% or less of the studies in this subset, such as neutropenia (4/16), cardiovascular diseases (3/16), pulmonary diseases (3/16), urinary catheter (3/16), chemotherapy (2/16), and renal insufficiency (2/16). The heterogeneity of the clinical background of the patients and the gap of the host epidemiological factors in these studies excluded further analysis between *Candida*-related biofilm isolates and clinical history.

## Mortality among patients with *Candida* biofilm

Further subgroup analysis using a random-effect model was realized to differentiate the *Candida*-related mortality rates between bloodstream infections with planktonic cells and biofilm formation. From the initial data set, only 15 studies evaluated the mortality among patients with *Candida*-related bloodstream infections. As shown in Table 3, the pooled mortality rate due to *Candida*-related bloodstream infections was 37.9% (95% CI: 26.2–50.2) of which the mortality associated with biofilm-forming infections was 70.0% (95% CI: 52.8–84.8).

In both scenarios, the mortality rate was statistically incremented among hospitalized patients ($p < 0.0001$). However, biofilm-related infections evidenced almost the double value of mortality rate in patients, when compared to all *Candida*-related bloodstream infections.

## Geographical distribution of biofilm-forming *Candida* spp. isolates

The prevalence rate of biofilm-related infections significantly varied among studies of different countries and regions. Therefore, a subgroup analysis was realized between the biofilm formation rates and the geographical region to evaluate possible statistically significant differences (Table 4). Subgroup analysis evaluated the biofilm prevalence between regions and countries with a minimum of published studies, at least two and three studies per region and country, respectively. However, Egger's test was not applied due to the low number of studies in this analysis.

Although the biofilm prevalence varied among regions, there were no statistically significant differences ($p = 0.4049$). Europe reported a greater number of studies and showed an intermediate biofilm prevalence among *Candida* spp. infections. Meanwhile, when comparing prevalence rates between countries, a statistically significant value was obtained ($p = 0.0074$). In the pairwise comparison analyses, Spain was significantly superior to Brazil ($p < 0.0001$), Italy ($p = 0.0263$), and India ($p = 0.0030$).

**Table 2. The reported clinical background of the patients with *Candida*-related bloodstream infections in the study set.**

| Study set | Total | Biofilm | Mortality | Mortality-related biofilm | Adult clinical conditions | | | | | | | | | | | | | | | | | | | | | | | | Pediatric clinical conditions | | | |
| | | | | | CA | IT | MV | CD | Neu | ND | CO | PD | GI | QMT | DI | AL | CRF | UC | CVC | RI | NGT | TPN | GAD | HIV | ANF | ANT | SC | ICU | PCVC | PVC | PB | LWB |
|---|---|---|---|---|---|---|---|---|---|---|---|---|---|---|---|---|---|---|---|---|---|---|---|---|---|---|---|---|---|---|---|---|
| Stojanovic et al., 2015 | 8 | 7 | 0 | 0 | 4 | 4 | NR | NR | NR | NR | NR | NR | NR | NR | 5 | 2 | 3 | NR | 6 | NR | 4 | 5 | NR | NR | NR | 6 | 4 | 6 | NR | NR | NR | NR |
| Banerjee et al., 2015 | 80 | 31 | 16 | 5 | 11 | NR | 9 | 5 | 6 | 6 | 7 | 11 | 19 | NR | 17 | 16 | 13 | 27 | 58 | 28 | NR | NR | NR | 1 | NR | 42 | 9 | NR | 0 | 19 | 14 | 13 |
| Guembe et al., 2014 | 54 | 45 | 0 | 0 | 16 | NR | NR | 6 | 6 | 6 | NR | NR | 6 | NR | NR | NR | NR | NR | 23 | NR | NR | NR | NR | NR | NR | NR | NR | NR | NR | NR | NR | 10 |
| Pongracz et al., 2016 | 93 | 43 | 43 | 23 | 25 | 19 | NR | NR | NR | NR | NR | NR | NR | NR | 20 | NR | NR | NR | NR | NR | NR | 22 | NR | 11 | NR | NR | 51 | NR | NR | NR | NR | NR |
| Vitalis et al., 2020 | 127 | 127 | 70 | 70 | 28 | 13 | 87 | NR | NR | NR | NR | NR | NR | NR | 41 | NR | NR | NR | NR | NR | NR | 68 | NR | 13 | 162 | 91 | 8 | 100 | NR | NR | NR | NR |
| Kumar et al., 2006 | 36 | 30 | 0 | 0 | 35 | NR | NR | NR | NR | NR | NR | NR | NR | NR | NR | NR | NR | NR | NR | NR | NR | NR | NR | 1 | NR | NR | NR | NR | NR | NR | NR | NR |
| Tumbarello et al., 2012 | 207 | 84 | 82 | 43 | 42 | 16 | NR | NR | NR | NR | 29 | 17 | 9 | NR | NR | NR | 21 | NR | 56 | NR | 27 | 58 | NR | 1 | NR | 75 | 38 | NR | NR | NR | NR | NR |
| Tumbarello et al., 2007 | 294 | 80 | 154 | 56 | 88 | 82 | NR | NR | 10 | NR | NR | NR | 16 | NR | NR | NR | NR | 136 | 30 | NR | NR | 72 | NR | NR | NR | NR | 100 | 57 | NR | NR | NR | NR |
| Marcos-Zambrano et al., 2017 | 22 | 22 | 0 | 0 | 21 | 13 | NR | NR | 4 | NR | NR | NR | 1 | NR | 76 | NR | 4 | NR | 19 | NR | NR | 13 | NR | 1 | 7 | NR | 4 | 2 | NR | NR | NR | NR |
| Tortonaro et al., 2013 | 451 | 160 | 13 | 11 | 136 | NR | NR | NR | NR | NR | NR | NR | NR | NR | NR | NR | NR | NR | NR | NR | NR | NR | NR | NR | NR | NR | 219 | 158 | NR | NR | 17 | NR |
| Muñoz et al., 2018 | 280 | 280 | 0 | 95 | 151 | 22 | 50 | 91 | 18 | 70 | 78 | 59 | NR | 53 | 69 | NR | 61 | NR | 201 | NR | NR | 152 | NR | 6 | 62 | 253 | 136 | 28 | NR | NR | NR | NR |
| Soldini et al., 2017 | 190 | 190 | 89 | 89 | NR | NR | NR | NR | NR | NR | NR | NR | NR | NR | NR | NR | NR | NR | 152 | NR | NR | 132 | NR | NR | NR | 177 | NR | 28 | NR | NR | NR | NR |
| Tascini et al., 2018 | 89 | 57 | 42 | 25 | NR | NR | 24 | NR | NR | NR | 13 | NR | NR | 7 | NR | NR | NR | 47 | 80 | 1 | 25 | 62 | NR | 13 | 75 | 53 | 35 | NR | NR | NR | NR | NR |
| Treviño-Rangel et al., 2018 | 89 | 89 | 32 | 32 | NR | NR | NR | NR | NR | NR | NR | NR | NR | NR | NR | NR | NR | 37 | 50 | NR | NR | 30 | NR | NR | 30 | 53 | 38 | NR | NR | NR | 4 | NR |
| Shin et al., 2002 | 101 | 58 | 0 | 0 | NR | NR | NR | NR | NR | NR | NR | NR | NR | NR | NR | NR | NR | NR | 41 | NR | NR | 35 | NR | NR | NR | NR | NR | NR | NR | NR | NR | NR |
| Atalay et al., 2015 | 50 | 8 | 0 | 0 | NR | NR | NR | NR | NR | NR | NR | NR | NR | NR | NR | NR | NR | NR | 18 | NR | NR | NR | NR | NR | NR | NR | NR | NR | NR | NR | NR | NR |
| Gangneux et al., 2018 | 319 | 181 | 105 | 55 | NR | NR | NR | NR | NR | NR | NR | NR | NR | NR | NR | NR | NR | NR | NR | NR | NR | NR | NR | NR | NR | NR | NR | NR | NR | NR | NR | NR |
| Herek et al., 2019 | 13 | 13 | 0 | 0 | NR | NR | NR | NR | NR | NR | NR | NR | NR | NR | NR | NR | NR | NR | NR | NR | NR | NR | NR | NR | NR | NR | NR | NR | NR | NR | NR | NR |
| Marcos-Zambrano et al., 2014 | 564 | 564 | 0 | 0 | NR | NR | NR | NR | NR | NR | NR | NR | NR | NR | NR | NR | NR | NR | NR | NR | NR | NR | NR | NR | NR | NR | NR | NR | NR | NR | NR | NR |
| Pannanusorn et al., 2012 | 393 | 231 | 0 | 0 | NR | NR | NR | NR | NR | NR | NR | NR | NR | NR | NR | NR | NR | NR | NR | NR | NR | NR | NR | NR | NR | NR | NR | NR | NR | NR | NR | NR |
| Pfaller et al., 1995 | 17 | 13 | 0 | 0 | NR | NR | NR | NR | NR | NR | NR | NR | NR | NR | NR | NR | NR | NR | NR | NR | NR | NR | NR | NR | NR | NR | NR | NR | NR | NR | NR | NR |
| Pham et al., 2019 | 46 | 38 | 23 | 13 | NR | NR | NR | NR | NR | NR | NR | NR | NR | NR | NR | NR | NR | NR | NR | NR | NR | NR | NR | NR | NR | NR | NR | NR | NR | NR | NR | NR |
| Prigitano et al., 2013 | 297 | 297 | 130 | 65 | NR | NR | NR | NR | NR | NR | NR | NR | NR | NR | NR | NR | NR | NR | NR | NR | NR | NR | NR | NR | NR | NR | NR | NR | NR | NR | NR | NR |
| Rajendran et al., 2016 | 280 | 245 | 0 | 0 | NR | NR | NR | NR | NR | NR | NR | NR | 121 | 30 | 153 | 128 | NR | NR | NR | 118 | NR | 123 | 133 | NR | 119 | NR | 40 | 128 | NR | NR | NR | NR |

(*Continued*)

**Table 2.** (Continued)

| Study set | Total | Biofilm | Mortality | Mortality-related biofilm | Adult clinical conditions | | | | | | | | | | | | | | | | | | | | | | | | Pediatric clinical conditions | | | |
|---|---|---|---|---|---|---|---|---|---|---|---|---|---|---|---|---|---|---|---|---|---|---|---|---|---|---|---|---|---|---|---|---|---|
| | | | | | CA | IT | MV | CD | Neu | ND | CO | PD | GI | QMT | DI | AL | CRF | UC | CVC | RI | NGT | TPN | GAD | HIV | ANF | ANT | SC | ICU | PCVC | PVC | PB | LWB |
| **Rodrigues et al., 2019** | 28 | 15 | 13 | 6 | NR | NR | NR | NR | NR | NR | NR | NR | NR | NR | NR | NR | NR | NR | NR | NR | NR | NR | NR | NR | NR | NR | NR | NR | NR | NR | NR | NR |
| **Sida et al., 2015** | 4 | 2 | 0 | 0 | NR | NR | NR | NR | NR | NR | NR | NR | NR | NR | NR | NR | NR | NR | NR | NR | NR | NR | NR | NR | NR | NR | NR | NR | NR | NR | NR | NR |
| **Thomaz et al., 2019** | 38 | 38 | 0 | 0 | NR | NR | NR | NR | NR | NR | NR | NR | NR | NR | NR | NR | NR | NR | NR | NR | NR | NR | NR | NR | NR | NR | NR | NR | NR | NR | NR | NR |
| **Tobudic et al., 2011** | 47 | 34 | 25 | 18 | NR | NR | NR | NR | NR | NR | NR | NR | NR | NR | NR | NR | NR | NR | NR | NR | NR | NR | NR | NR | NR | NR | NR | NR | NR | NR | NR | NR |
| **Tulasidas et al., 2018** | 74 | 55 | 0 | 0 | NR | NR | NR | NR | NR | NR | NR | NR | NR | NR | NR | NR | NR | NR | NR | NR | NR | NR | NR | NR | NR | NR | NR | NR | NR | NR | NR | NR |
| **Tulyaprawat et al., 2020** | 48 | 45 | 0 | 0 | NR | NR | NR | NR | NR | NR | NR | NR | NR | NR | NR | NR | NR | NR | NR | NR | NR | NR | NR | NR | NR | NR | NR | NR | NR | NR | NR | NR |
| **Turan et al., 2018** | 162 | 145 | 0 | 0 | NR | NR | NR | NR | NR | NR | NR | NR | NR | NR | NR | NR | NR | NR | NR | NR | NR | NR | NR | NR | NR | NR | NR | NR | NR | NR | NR | NR |

CA: malignancy; IT: Immunosuppressive Therapy; MV: Mechanical Ventilation; CD: Cardiovascular Disease; Neu: Neutropenia; ND: Neurological Disorders, CO: Corticoids; PD: Pulmonary Disorders; GI: Gastro Intestinal and Hepatically Disease; QMT: Chemotherapy; DI: Diabetes; AL: Alcoholism; CRF: Chronic Renal Failure; UC; Urinary Catheter; CVC: Central Venous Catheter; RI: Renal Insufficiency; NGT: Nasogastric Tube, TPN: Total Parenteral Nutrition; GAD: Genetic Autoimmune Disorders; HIV: Human Immunodeficiency Virus; ANF: Prior Antifungal Therapy; ANT: Prior Antibacterial Therapy; SC: Surgical conditions; ICU: Intensive Care Unit; PCVC: Pediatric Central Venus Catheter; PVC: Peripheric Venus Catheter; PB: Preterm Bird; LBW: Low Weight Bird; NR: Not Reported in the study.

**Table 3. Pooled mortality rates in bloodstream infections due to *Candida* spp.**

| | k | Mortality rate (95% CI) (%) | Random model | | | |
|---|---|---|---|---|---|---|
| | | | Q | I² | τ | *p* |
| *All Candida* spp. bloodstream infections | 15 | 37.9 (26.2–50.2) | 493.82 | 97.2 | 0.237 | < 0.0001 |
| Biofilm-forming | 15 | 70.0 (52.8–84.8) | 345.47 | 95.9 | 0.331 | < 0.0001 |

k, Number of studies; Q, I² and τ, Heterogeneity indexes; *p*, Random effect model significance level. Mortality rates were estimated within 30 days after diagnosis and confirmation of *Candida* spp. bloodstream infection. The studies considered (k = 15) were those in which a sample corresponded to an individual and reported deaths related to biofilm-formers strains.

## Biofilm-forming capability in *Candida* spp. isolates

*Candida* spp. isolates vary in their ability to form biofilms, being usually categorized as low (LBF), intermediate (IBF), and high biofilm formers (HBF) according to biomass production (S1–S3 Figs). Briefly, biofilm forming capacity was assessed using the crystal violet or XTT assays, measuring the biofilm mass. *Candida* isolates were cultured in 96-well plates at 37˚C for 24 h and the biomass of each isolate was measured. Then, isolates were grouped based on their level of biomass, more exactly: low biofilm formers (LBF) showed a biomass production below the 1st quartile (Q₁; $Abs_{isolate} < 0.432$), intermediate biofilm formers (IBF) evidenced a biomass production in the 2nd quartile (Q₂; $0.432 < Abs_{isolate} < 1.07$), and high biofilm formers (HBF) demonstrated a biomass production higher the 1st quartile 3rd quartile (Q₃; $Abs_{isolate} > 1.07$), as previously described by Monfredini et al. [16] and Vitális et al. [17]. Eighteen studies reported this biofilm classification and so a subgroup analysis was realized (Table 5).

Statistically significant differences were found among *Candida* isolates according to their biofilm-forming capability ($p < 0.0001$), evidencing a low number of *Candida* isolates related to intermediate biofilms. No publication bias was detected in both subgroups according to Egger's linear regression test.

## Evaluation of biofilm formation between different *Candida* species

Although *Candida* spp. isolates vary in their ability to form biofilms, little is known about this biofilm-forming ability among *Candida* species. Each category of biofilm was further evaluated among *Candida* species to evaluate the most virulent *Candida* species (S1 Table). When

**Table 4. Subgroup analysis for different geographical regions and countries.**

| Subgroups | k | Prevalence (95% CI) (%) | Random model | | | |
|---|---|---|---|---|---|---|
| **Region** | | | Q | I² | τ | *p*\* |
| **Europe** | 17 | 81.0 (63.3–94.0) | 2267.21 | 99.3 | 0.407 | 0.4049 |
| **Asia** | 9 | 67.9 (48.1–85.0) | 171.49 | 95.3 | 0.283 | |
| **South America** | 3 | 91.6 (50.7–100.0) | 31.83 | 93.7 | 0.387 | |
| **North America** | 2 | 94.0 (55.1–100.0) | 12.94 | 92.3 | 0.319 | |
| **Country (≥3 studies)** | | | | | | |
| **Italy** | 6 | 69.1 (32.0–95.8) | 1095.33 | 99.5 | 0.471 | 0.0074 |
| **India** | 5 | 72.3 (46.2–92.7) | 55.54 | 92.8 | 0.267 | |
| **Spain** | 4 | 98.9 (93.5–100.0) | 33.85 | 91.1 | 0.126 | |
| **Brazil** | 3 | 91.6 (50.7–100.0) | 31.83 | 93.7 | 0.387 | |

k, Number of studies; Q, I² and τ, Heterogeneity indexes; *p*\*, Significance level in subgroup analysis.

**Table 5. Overall effects in subgroups based on biofilm-forming capability.**

| Biofilm-forming capability | k | Prevalence (95% CI) (%) | Egger's test | Random model | | | |
|---|---|---|---|---|---|---|---|
| | | | $p$ | Q | $I^2$ | $\tau$ | $p^*$ |
| **High (HBF)** | 18 | 35.0 (26.6–43.9) | 0.768 | 313.94 | 94.58 | 0.177 | < 0.0001 |
| **Intermediate (IBF)** | 18 | 18.9 (7.8–33.1) | 0.457 | 1074.52 | 98.42 | 0.334 | < 0.0001 |
| **Low (LBF)** | 18 | 36.2 (24.7–48.5) | 0.370 | 623.25 | 97.27 | 0.253 | < 0.0001 |

k, Number of studies; Q, $I^2$ and $\tau$, Heterogeneity indexes; $p^*$, Random effect model significance level in subgroup analysis. The selected studies (k = 18) categorized the strains according to their biofilm-forming capability using only methods based on biomass quantification through spectrophotometric measures.

analyzing HBF (Table 6), *C. tropicalis* was the most prevalent HBF overpassing *C. albicans* and *C. parapsilosis* by a factor of 2. More precisely, the HBF prevalence of *C. tropicalis* was the highest showing statistically significant differences with the other *Candida* species, except for *C. krusei* ($p = 0.5477$) and *C. glabrata* ($p = 0.0896$).

In order to comprehend how these two major factors: countries and *Candida* species could actually explain the high heterogeneity showed in our data, we carried out a meta-regression analysis. The inclusion of both variables as interacting variables in a multiplicative model ($R^2 = 59.13\%$, $p < 0.0001$) explained more than an additive model ($R^2 = 43.48\%$, $p < 0.0001$), regarding the prevalence of biofilm formation.

## Evaluation of antifungal resistance pattern among *Candida* isolates

Multiple antifungal resistance among candidiasis has become a serious public health issue, leading to clinical complications and expensive costs. A subgroup analysis based on antifungal resistance was also realized among our study set. Due to the different methodologies used to test susceptibility, the number of studies not enough to analyze statistically antifungal resistance rates between *Candida* species. As shown in Table 7, the rates of antifungal resistance to fluconazole, voriconazole, and caspofungin related to biofilm-forming strains were 70.5, 67.9, and 72.8%, respectively.

**Table 6. Subgroup analysis between different *Candida* species.**

| Species | k | BF strains (n) | Prevalence of HBF % (95% CI) | Random model | | | |
|---|---|---|---|---|---|---|---|
| | | | | Q | $I^2$ | $\tau$ | $p^*$ |
| ***C. albicans*** | 22 | 1461 | 30.3 (20.5–41.0) | 225.66 | 95.6 | 0.173 | 0.0454[a] |
| **non-albicans *Candida* species** | 26 | 1868 | 43.6 (34.5–52.9) | 306.69 | 87.6 | 0.230 | |
| *C. albicans* | 22 | 1461 | 30.3 (20.5–41.0) | 225.66 | 95.6 | 0.173 | |
| *C. glabrata* | 17 | 387 | 37.6 (0.1–71.0) | 95.0 | 95.8 | 0.325 | < 0.0001[b] |
| *C. tropicalis* | 17 | 331 | 67.5 (58.3–76.3) | 11.71 | 31.7 | 0.069 | |
| *C. parapsilosis* | 20 | 744 | 29.6 (20.3–39.9) | 69.9 | 84.3 | 0.154 | |
| *C. krusei* | 10 | 68 | 52.8 (0.1–94.9) | 30.12 | 83.4 | 0.409 | |
| ** **Other species** | 20 | 338 | 40.7 (26.5–55.6) | 22.49 | 60.0 | 0.139 | |

k, Number of studies; Q, $I^2$ and $\tau$, Heterogeneity indexes; $p^*$, Random effect model significance level in subgroup analysis.

[a] Comparison between *C. albicans* and non-albicans *Candida* species.

[b] Comparison between all *Candida* species.

** Other species includes *C. dublinensis* (n = 12), *C. quilliermondi* (n = 25), *C. lusitaniae* (n = 10), *C. haemulonii* (n = 4), *C. lypolitica* (n = 1), *C. pelliculosa* (n = 1) and unreported species (n = 285).

**Table 7. Summary of subgroup analysis for antifungal resistance in *Candida* spp. isolates.**

| Studies | k | Antifungal resistance rate % (95% CI) | | |
|---|---|---|---|---|
| | | Fluconazole | Voriconazole | Caspofungin |
| Mixed/Planktonic cells | 3 | 15.1 (0.7–41.2) | 1.6 (0.1–4.4) | 3.1 (0.0–20.76) |
| Biofilm-forming strains | 2 | 70.5 (54.6–84.5) | 67.9 (51.8–82.3) | 72.8 (55.1–87.8) |
| Cochran's Q* | | 11.68 | 85.15 | 22.88 |
| *p*-value** | | 0.0006 | < 0.0001 | < 0.0001 |
| Not reported/ Other methods | 26 | - | - | - |

k, Number of studies; Q*, Test of heterogeneity between groups; *p*\*\*, Random effect model significance level in subgroup analysis. Subgroup analysis based on antifungal resistance contains k = 5 studies. Egger's test may lack the statistical power to detect bias when the number of studies is small (i.e., k < 10).

When comparing to planktonic cells, all *Candida*-related biofilm isolates showed a statistical increment of resistance against the three antifungals evaluated in the study ($p < 0.001$).

## Discussion

The present study evaluated a possible relationship between *Candida*-related biofilm formation, bloodstream infections, and mortality among hospitalized patients. Invasive mycoses are responsible every year for more than two million infections worldwide and for, at least, as many deaths as tuberculosis or malaria. Candidiasis, aspergillosis, cryptococcosis, and pneumocystosis cause more than 90% of reported deaths associated with invasive mycoses [18]. Among them, the most frequent mycosis is invasive candidiasis causing high morbidity in critically ill patients [19].

### Overall effects of *Candida* biofilms in infections and mortality

As previously referred, around 70.0% of candidemia reports were caused by biofilm-forming strains. However, its biofilm formation was less than in isolates from urogenital infections [20–23] and even respiratory tract infections [22, 23]. Still, the rate of candidemia-associated biofilm infections was higher than oral-related biofilm infections [24] and more than invasive infections [25]. These findings are in agreement with the Institute of Health in the United States, which estimates that biofilms are responsible, in one way or another, for over 80% of all microbial infections [12]. Yet, the reports of *Candida*-associated biofilm infections varied greatly between published studies possibly due to the lack of differentiation between *Candida* species, the experience of the researchers, the number of *Candida* isolates in the study set, and the diversity of biofilm detection and quantification methodologies and its subsequent classification within the study set, such as crystal violet assay, biomass measure, XTT reduction assay, and microtiter plate method [8, 12].

Another issue concerns the lack of differentiation between planktonic and biofilm-related *Candida* infections in the diagnosis of the clinical laboratories at public health system [19, 26]. The traditional clinical microbiology laboratories have focused on testing planktonically isolated microorganisms and reporting the susceptibility to various antimicrobials under planktonic growth conditions [27]. While the authors from the studies of this meta-analysis applied a further analysis by evaluating the ability of biofilm production in *Candida* isolates through an *in vitro* biofilm assay. In *Candida* biofilms, traditional techniques require device removal followed by culture or microscopy of a catheter segment, while catheter-sparing diagnostic tests include paired quantitative blood cultures. However, as previously indicated by Høiby et al. (2015) and Bouza et al. (2013), the number of positive peripheral blood cultures also

seems to be a promising diagnostic tool to diagnose catheter-related candidemia without directly removing the catheter [27, 28]. Therefore, an implementation of a new gold standard methodology is vital to a better characterization of microbial-associated infections avoiding unproductive treatments among hospitalized patients. The mortality rate caused by biofilm formation in *Candida*-related infections was almost double when compared to planktonic infections. Other studies already stated the burden of invasive candidiasis and its severe outcomes [1, 29], indicating biofilm formation and antifungal resistance as main risk factors among patients. Moreover, we report a pooled attributable mortality of 37.9% to *Candida*-related bloodstream infection with planktonic cells, which is in agreement with previous reports [1, 18, 30, 31]. These studies reported a mortality range between 25 and 40%, showing a higher mortality incidence among ICU or burn patients, and immunocompromised patients [32]. While the mortality associated with biofilm-forming strains was 70.0% in *Candida*-related bloodstream infections. However, this correlation has been debated by several authors [10, 16, 33, 34], reporting different mortality rates (25–70%).

It is also important to mention that the ability to quicky proliferate and to establish biofilm is not exclusively dependent of the type of *Candida* species and even strains in a blood-related infection, but it is also influenced by their interaction with host homeostasis and variations (mucosal pH shifts or nutritional changes), previous use of antibiotics, and immune system alterations (such as secondary effect of stress or immunosuppressant therapy) [35].

The $I^2$ observed in the forest plot indicate a high heterogenic data. The $I^2$ is a measurement of the heterogeneity that is not caused by variations in the sample size considered in each study. Therefore, this high value and also the geometry of the funnel plot indicates the possibility of major sources of variation across the studies. Some of the sources of variations can clearly be related with the differences previously described (i.e., methodology, *Candida* species, etc.) and consequently the pooled effect around the 80% need to be considered with caution. Several factors can be modulating this pooled effect leading to higher and/or lower values. In this context, the present meta-analysis was unable to study any correlations between clinical or epidemiological factors and mortality in patients with biofilm-related blood infections. These heterogeneity and gaps on the selected studies constitute the main drawback of our study. However, it is also well-known that the ability to establish biofilms among *Candida* species is an important virulence factor contributing to a more severe infection in patients [36] and it is worth to be studied. The observed heterogeneity was the leading cause to consider the effect of several variables like geographical distribution and *Candida* species. However, the missing information in the consulted scientific literature can be an important source of unexplained variation.

## Geographical distribution of *Candida* biofilm-related infections

World incidence of invasive candidiasis is difficult to estimate because the criteria used for diagnosing and categorizing invasive candidiasis are quite different [6, 8, 9]. Also, most studies restricted many factors in their group set, such as the range age of patients and their health status. The present meta-analysis recollected data from diverse study sets demonstrating the *Candida*-related biofilm infections as a main nosocomial infection, but only 16 of 31 studies partially reported the clinical background of the patients (Table 2), such as patients suffering from immunodeficiency, receiving organ transplantations, under major surgery, or treated with cancer chemotherapy and different primary hospitalizations, and no epidemiological factors were available. Only a study realized in a tertiary care hospital of southern India reported the clinical backgrounds in adult and pediatric patients [37], evidencing central venous catheter and low weight at birth as the most prevalent risk factors in these population sets, respectively.

Generally, the number of patients in surveillance studies is very low and there are many gaps in our knowledge on the true epidemiology of invasive candidiasis in many regions of the world [19]. As expected, around 55% of our data set belonged to European studies (17/31), where the rate of biofilm-related infections varied greatly among countries showing Spain with statistical differences in the incidence of *Candida*-related biofilm infections in hospitalized patients in comparison with other countries. However, Cesta and colleagues recently reported Italy as the one region with a higher number of deaths caused by antibiotic-resistant bacteria and biofilm-related infections [38]. Due to European Centre for Disease Prevention and Control (ECDC) reported a spread of multi-drug resistant strains (MDR) in Italy, in particular of the bacterial species of *Pseudomonas aeruginosa*, *Klebsiella pneumoniae*, and *Acinetobacter baumannii* [38], it is plausible that the *Candida*-related biofilm incidence among hospitalized patients in Italy had been underrated. Likewise, only two and three studies in our data set belong to North and South America, respectively. All three studies of South America were indeed from Brazil, demonstrating one of the highest *Candida*-related biofilm incidences among hospitalized patients (91.6%). However, no further information was available in the remaining Latin-American countries with the criteria selection of the present meta-analysis.

We can notice in the meta-analysis that the values of $I^2$, Q and other indicators also suggest a high heterogeneity within each group. It is an indicator that other factors can be involved. For example, if we consider only the articles from Italy, we can notice that the sample size in 5 of 6 studies do not considerably differ but the effect size is quite different (this will impact directly in the funnel plot geometry as presented in Fig 3). In three studies, we found a low prevalence of biofilm formation [33, 39, 40] while in other two articles we found a high prevalence of biofilm formation [41, 42]. This distribution suggests that factors quite beyond the geography are possible causes of heterogeneity within groups.

## Association between different *Candida* species in biofilm and infections

The number of *Candida* species with clinical importance in humans is relatively small, more exactly, *Candida albicans*, *Candida glabrata*, *Candida tropicalis*, *Candida parapsilosis*, and *Candida dubliniensis* [43]. *C. albicans* is the most reported *Candida* species worldwide in different ethnic populations [34, 44–47], being responsible for the majority of oral and systemic candidiasis cases. However, there has been an increase in the number of reports about non-albicans *Candida* infection in the last years and even surpassing *C. albicans* in terms of incidence and attributable mortality [25, 31, 34, 42, 48–51]. This new scenario could be attributed to the implementation of better molecular techniques in the identification of *Candida* species [21, 29, 52].

Our results demonstrated *C. tropicalis* as the most prevalent HBF evidencing statistical dominance among *Candida* species. Although *C. tropicalis* is described as a species with normal to high biofilm-forming capacity [36], it is commonly related to infections in prosthetic joints, endodontic issues, ulcerative colitis [53–55]. *C. tropicalis* biofilm is characterized by chains of cells with thin, but large, amounts of extracellular matrix material with low sums of carbohydrate and protein [36, 40]. Furthermore, Silva and colleagues showed that matrix material extracted from biofilms of *C. tropicalis* and *C. albicans* contained carbohydrates, proteins, hexosamine, phosphorus and uronic acid [55]. However, hexosamine was the major component quantified in *C. tropicalis* biofilm (27%). *C. tropicalis* biofilms are described as a dense network of yeast cells with evident different filamentous morphologies [36].

After *C. tropicalis*, the present meta-analysis showed *C. krusei* and *C. glabrata* as the second and third most prevalent HBF among *Candida* species, more exactly, 52.8 and 37.6%, respectively. *C. krusei* is characterized by a thick multilayered biofilm of pseudohyphal forms

embedded within the polymer matrix [56], being categorized with a high ability to establish biofilm [36]. Several mucosal infections and pneumonia are caused by *C. krusei* [23, 56]. Although *C. glabrata* is known to develop less biofilm, it is characterized to produce high content of both protein and carbohydrate [40, 57]. *C. glabrata* is commonly associated with infections among patients with total parenteral nutrition, periodontal disease, ventilator-associated and non-healing surgical wounds [58]. *C. glabrata* biofilms are structured on multilayers of blastospores with high cohesion among them [55]. The elucidation of these biofilm-forming abilities and properties among *Candida* species could provide a promising step toward the improvement of treatments.

Until this point, we have showed that *Candida* species and geographical distribution can be related with our data heterogeneity. The actual combination of both variables in a multiple meta-regression model as interacting variables explained more than the 50% of the global variability. The lack of clinical information and many other discussed variables are probably related, at least partially, with the remained variability. Unfortunately, as previously explained, this information is not accessible for most of the studies and constitute by itself a recommendation in further studies.

### Antifungal resistance among *Candida*-related biofilm infections

*Candida* spp. infections had successfully become more difficult to treat in the last decade due to the growth of immunogenic diseases, the disproportionate use of immunosuppressive drugs, malnutrition, endocrine disorders, the widespread use of indwelling medical devices, broad-spectrum antibiotics, aging, and an increase of the number of patients among the population [36, 59]. Thus, the morbidity and mortality associated with candidiasis are still very high, even using the actual antifungal drugs [59]. The main antifungal drugs applied to *Candida* infections are azoles, polyenes, and echinocandins [60]. Briefly, azoles (such as fluconazole and voriconazole) block ergosterol synthesis by targeting the enzyme lanosterol 14α-demethylase and leading to an accumulation of toxic sterol pathway intermediates. While echinocandins (such as caspofungin) aim for the synthesis of 1,3-β-glucan (a cell wall component), being the ideal antifungal drug of choice in severe cases of candidemia [61, 62]. As previously referred, the rates of antifungal resistance to fluconazole, caspofungin, and voriconazole in biofilm cells surpassed planktonic cells by a factor of 4.7, 23.5, and 42.4, respectively. Despite the number of studies comparing resistance between planktonic and biofilm cells among *Candida* species is still scarce, these results are in agreement with the literature postulations [36, 63]. Numerous reasons are attributed to this enormous resistance against antifungal drugs in *Candida*-related biofilms, such as high cell density, growth rate reduction, nutrient limitation, matrix extracellular production, presence of persister (dormant and non-dividing) cells, phenotypic shift, and high sterols content on membrane cell [36, 59, 63]. So, the treatment for *Candida*-biofilm infections requires a comprehensive knowledge of the complex mechanisms underlying the interaction between a biofilm and its host.

Although no efficient treatment for *Candida* biofilms has been found yet, several promising strategies are being explored. New therapeutic targets, such as the genes involved in biofilm development and the quorum-sensing systems, are considered an alternative treatment to the currently antifungal drugs.

## Conclusions

In summary, several studies on the prevalence of *Candida* biofilms in bloodstream infections have been published across the world, allowing some conclusions on its mortality, species, and virulence in different geographic regions. However, a lot of information is missing, such as the

lack of a thorough clinical background from the patients and the diversity of the primary infections from the patients. Further studies are needed to close gaps in our understanding of the incidence of *Candida* biofilms and to monitor trends in antifungal resistance and species shifts.

To the authors' best knowledge, this meta-analysis is one of the few that explored the association of biofilm production among different *Candida* species in bloodstream infections [64–67], using data published worldwide and adhering to the Preferred Reporting Items for Systematic Reviews and Meta-Analyses guideline. Although the present meta-analysis was performed methodically, there are some limitations of this study: (1) heterogeneity exists in some subgroup and overall analyses; (2) relationship between mortality and each *Candida*-related biofilm species could not be assessed; and, (3) a detailed analysis of antifungal resistance in *Candida* biofilms was not possible. These limitations are due to a lack of sufficient published data. Therefore, early detection of biofilms and a better characterization of *Candida* spp. bloodstream infections should be considered, which eventually will help preserve public health resources and ultimately diminish mortality among patients.

## Materials and methods

### Data selection, search strategy, and study guidelines

This study was conducted following Preferred Reporting Items for Systematic Reviews and Meta-Analyses (PRISMA) strategies (S1 File) [68]. Web of Science, Scopus, PubMed, and Google Scholar databases were searched for English papers using the following medical subject heading terms (MESH): "invasive candidiasis"; "bloodstream infections"; "biofilm formation"; "biofilm-related infections"; "mortality"; and, "prevalence".

In each electronic database, a combination of MESH terms was used to conduct the search applying the following strategy (in the MEDLINE for example): "(Candida) AND (biofilm [Title/Abstract]) AND (mortality)." All studies published until 30th July of 2020 were retrieved. The articles reporting the prevalence of bloodstream infections biofilm-related, the mortality rates, and the species identification of *Candida* isolates were included. The references of these articles were also checked for finding additional records. The data selection was limited to human clinical isolates and studies in English. All references were compiled into a database Zotero Library and then managed using Excel.

### Screening process

Duplicates were initially identified and eliminated in Zotero after entering all the recognized studies into an Excel self-created database (S2 File). All articles were assessed by two reviewers (MBA-C and FSC-M) by screening titles, abstracts, topics, and finally full texts. At each level, the reviewers independently screened the articles and finally merged their conclusions. An additional examination of the selected articles was realized by a third author (AM) focused on the homogeneity of the eligibility criteria of previous reviewers in the initial data set. Discrepancies were resolved by discussion before finalizing the records for the evaluation of eligibility criteria. In case of disagreements, the third assessor (AM) was assigned to make a final decision.

### Eligibility criteria

The major inclusion criteria included reporting the rate of biofilm formation and the prevalence of biofilm-related to *Candida* species, including observational studies (more exactly, cohort, retrospective, and case-control studies). Furthermore, data regarding the mortality

rate, the geographical location of the study set, and the use of anti-fungal agents in clinical isolates were also extracted from the studies.

All studies without information about biofilm formation or clinical *Candida* isolates were consequently excluded. The method to quantify biofilm biomass was not a criterion to include or exclude any paper in this meta-analysis. Concerning antifungal resistance rate, only studies that used the standard susceptibility tests according to the Clinical and Laboratory Standards Institute (CLSI) or European Committee on Antimicrobial Susceptibility Testing EUCAST were selected for the present study.

Reviews, editorials, congress or meeting abstracts, literature in languages other than English, case reports, and letters to editors were excluded from the final data set. Finally, articles without full text available, duplicate reports on different databases, and studies with unclear or missing data were also omitted.

## Data extraction and quality assessment

Methodological quality assessment of the studies was performed using a checklist for necessary items as outlined in the Critical Appraisal Skills Programmed checklists [69]. For each article, a series of critical questions were asked. If the pertinent data were presented, the question was scored "yes." If there was any doubt or no information in the study, that question was marked as "no". A data extraction form was designed to extract the relevant characteristics of each study (S1 and S2 Files). The extracted information included the first authors' names, time of the study, year of publication, location, sample size, biofilm formation rate, *Candida* species and its categorization (as *C. albicans* and non-*albicans Candida* species), the correlation between biofilm formation and antifungal resistance, and the type of biofilm. The type of biofilm was categorized as low biofilm formers (LBF), intermediate biofilm formers (IBF), and high biofilm formers (HBF). The initial two authors (MBA-C and FSC-M) extracted all data, further confirmation and final evaluation were realized by the lead authors (AM and ET).

## Data analysis and statistical methods

Meta-analysis was performed using several R packages ("meta" [70], "metafor" [71], "poibin" [72], and "stringr" [73]) of R version 3.4.3 [74] and RStudio version 1.3.1073 [75] (S3 File). The rate of biofilm formation was computed, and values were reported with confidence intervals (CI) of 95%. The heterogeneity was assessed by the Cochrane Q and $I^2$ tests. The $I^2$ metric indicates the amount of heterogeneity that is not related with sampling size variation. Moreover, it is also independent of the number of studies included in the meta-analysis (in contrast to the Cochrane Q metric). Considering the heterogeneity indices, the random-effects model was then used for meta-analysis of the selected studies, and the Freeman-Tukey transformation was also applied to calculate the pooled frequencies. To estimate the between-study variance in a random-effects model we use tau-squared, and its square root is the estimated standard deviation of underlying effects across studies. Subgroup analyses were conducted based on the type of biofilm, biofilm-related species, geographical regions, and antifungal resistance rates. Outliers' analysis was done with the Baujat diagram, while quantitative Egger weighted regression test and Funnel plot were used to evaluate the eventual existence of publication bias. In statistical analysis, *p*-values <0.05 were considered as significant statistical results. We used the multiple meta-regression analysis with the "metareg" function from "meta" to explore the contribution to model heterogeneity of several variables. In this approach, the maximum-likelihood method was used.

## Supporting information

**S1 Fig. Forest plot of the meta-analysis of the prevalence of high biofilm producers in *Candida* spp. isolates.**
(TIF)

**S2 Fig. Forest plot of the meta-analysis of the prevalence of intermediate biofilm producers in *Candida* spp. isolates.**
(TIF)

**S3 Fig. Forest plot of the meta-analysis of the prevalence of low biofilm producers in *Candida* spp. isolates.**
(TIF)

**S1 Table. Subgroup analysis between different *Candida* species and biofilm-forming capability.**
(DOCX)

**S1 File. The PRISMA statement for reporting meta-analysis of the present study.**
(DOCX)

**S2 File. The databases of the present study for metanalyses process.**
(XLSX)

**S3 File. The R-code used in the present meta-analysis.**
(TXT)

## Acknowledgments

A special recognition deserves all colleagues of the Microbiology Institute of USFQ and COCIBA, as well as the Research Office of Universidad San Francisco de Quito.

## Author Contributions

**Conceptualization:** María Belén Atiencia-Carrera, António Machado.

**Data curation:** María Belén Atiencia-Carrera, Fausto Sebastián Cabezas-Mera, António Machado.

**Formal analysis:** María Belén Atiencia-Carrera, Fausto Sebastián Cabezas-Mera, António Machado.

**Funding acquisition:** António Machado.

**Investigation:** María Belén Atiencia-Carrera, Fausto Sebastián Cabezas-Mera, António Machado.

**Methodology:** Fausto Sebastián Cabezas-Mera, Eduardo Tejera, António Machado.

**Project administration:** António Machado.

**Software:** Fausto Sebastián Cabezas-Mera, Eduardo Tejera.

**Supervision:** António Machado.

**Validation:** Eduardo Tejera, António Machado.

**Visualization:** Fausto Sebastián Cabezas-Mera.

**Writing – original draft:** María Belén Atiencia-Carrera, Fausto Sebastián Cabezas-Mera, António Machado.

**Writing – review & editing:** Eduardo Tejera, António Machado.

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
