## [Decision Letter · Decision Letter 0]

8 Nov 2021

PONE-D-21-23498Prevalence of Biofilms in Candida spp. bloodstream infections: A Meta-analysisPLOS ONE

Dear Dr. Antonio Machado

Thank you for submitting your manuscript to PLOS ONE. After careful consideration, we feel that it has merit but does not fully meet PLOS ONE’s publication criteria as it currently stands. Therefore, we invite you to submit a revised version of the manuscript that addresses the points raised during the review process.

Please submit your revised manuscript by  November 20, 2021. If you will need more time than this to complete your revisions, please reply to this message or contact the journal office at plosone@plos.org. Please include the following items when submitting your revised manuscript:A rebuttal letter that responds to each point raised by the academic editor and reviewer(s). You should upload this letter as a separate file labeled 'Response to Reviewers'.A marked-up copy of your manuscript that highlights changes made to the original version. You should upload this as a separate file labeled 'Revised Manuscript with Track Changes'.An unmarked version of your revised paper without tracked changes. You should upload this as a separate file labeled 'Manuscript'.

We look forward to receiving your revised manuscript.

Kind regards,

Surasak Saokaew, PharmD, PhD, BPHCP, FACP

Academic Editor

PLOS ONE

2. Please confirm that you have included all items recommended in the PRISMA checklist including:

- the full electronic search strategy used to identify studies with all search terms and limits for at least one database.

- a Supplemental file of the results of the individual components of the quality assessment, not just the overall score, for each study included.

- See https://journals.plos.org/plosmedicine/article?id=10.1371/journal.pmed.1000100#pmed-1000100-t003 for guidance on reporting.

4. Thank you for stating the following in the Funding Section of your manuscript:

“This work was supported by COCIBA Research Grant 2018-2019 through project ID: 12260 entitled “Adhesión inicial y resistencia antimicrobiana de Candida sp. aisladas de la microbiota humana”, under regulations of the Ministry of Health of Ecuador (Contrato Marco de Acceso a los Recursos Genéticos No. MAE-DNB-CM-2016-0046).”

“No - The funders had no role in study design, data collection and analysis, decision to publish, or preparation of the manuscript.”

Reviewers' comments:

Reviewer's Responses to Questions

**Comments to the Author**

1. Is the manuscript technically sound, and do the data support the conclusions?

Reviewer #1: No

Reviewer #2: Yes

2. Has the statistical analysis been performed appropriately and rigorously? 

Reviewer #1: Yes

Reviewer #2: No

3. Have the authors made all data underlying the findings in their manuscript fully available?

Reviewer #1: Yes

Reviewer #2: No

4. Is the manuscript presented in an intelligible fashion and written in standard English?

Reviewer #1: Yes

Reviewer #2: Yes

5. Review Comments to the Author

Reviewer #1: Although this paper has merit in concept the authors appear to have tried to undertake a review that requires clinical knowledge without the background to do so. The ability of an organism to produce biofilm under laboratory conditions does not necessarily represent the condition of the organism under its clinically infectious state, especially when the review is aimed at only blood cultures, rather than taking into account infections linked to indwelling surfaces such as heart valves and prosthetics.

The authors attempt to compare levels of biofilm production between papers but do not comment on methodology utilised to assess biofilm production to support the reader in understanding whether they are reviewing comparable methods. Discussion of clinical presentations include items such as neonates and TPN without linking the fact that pre-term infants are likely to be on TPN and therefore this is correlation not causation. Is some of the data linked to considerable variation in mortality linked to this lack of clinical interpretation or patient factors or items such as drug availability. Were lines left in situ as a continuous source for instance - without information on whether the clinical analysis is comparable it is difficult to say, as noted with the laboratory comparisons, if the data is truly comparable.

Therefore although the paper is of interest and could be modified for publication with further information I do not feel it is publishable in its current form.

Reviewer #2: The authors present a meta-analysis of Candida infections, and compare prevalence among geographic locations, Candida species, phenotypes (planktonic vs. biofilm; “biofilm forming capability”), and they also consider Candida resistance to a few anti-fungal agents.

There are some major issues, the most serious is that there are many confusing issues with the statistical diagnostics and reporting of statistical results. Secondly, the authors could provide a lot more details about how biofilm associated infections were diagnosed.

1. Statements re: standard statistical tests for publication bias could be better worded. It’s great that the authors include the funnel plot so that readers can decide for themselves whether the plot suggests publication bias. Asymmetry IS evident in the plot, it is just that Egger’s test fails to find that the asymmetry is drastic enough to suggest publication bias. In light of this, the statement in lines 24-25 about the funnel plot and Egger's test showing no publication bias is incorrect (p=0.896), replace with a statement that says that Egger's test failed to show publication bias, or failed to find publication bias. Similarly, also change the incorrect wording on line 113, “the symmetry of the funnel plot confirmed the hypothesis of absence of bias,” it did not.

2. Even more worrisome is that the funnel plot does not show a funnel at all, what does this mean?!?! Are the authors conclusions still valid?

3. For the PLOS ONE audience, please explain why the funnel plot should look like a funnel, and explain the axes.

4. Line 100, what does the p-value < 0.001 associate with? Not the t-stat of 0.387! The Q stat is the test statistic for the test of heterogeneity across studies. What is the t-stat for? Both these stats have associated p-values, but only one is reported here.

5. Tables 2 and 3, What are k, P, p*, Q, I and $\\tau$ in the table? Is k the number of studies? Is P the p-value for Egger's test? For the general PLOS audience not familiar with meta-analyses, please provide brief summaries of these.

6. Lines 149-150, “Although the biofilm prevalence varied among regions, no statistically significant value was obtained in this subgroup analysis” is uninformative. Consider replacing with “Although the biofilm prevalence varied among regions, there were no statistically significant differences (p = 0.4049).”

7. Lines 151-152, the authors are keen to point out getting a small p-value, “Meanwhile, when comparing prevalence rates between countries, a statistically significant value was obtained (p = 0.0074)”. The authors should make clear WHY there was small p-value. That p-value says there is some statistically significant difference between at least 2 countries, but the authors do not make clear which countries are statistically significantly different. In lines 153 and 154 the authors mention that Italy has the lowest rate and that Spain has the highest. Is Spain statistically significantly the highest than all other countries? Or maybe Spain is statistically significantly higher than just Italy? The authors make a nebulous reference to this comparison again in lines 236-237 which seems to suggest that Spain is statistically significantly higher than just Italy. Please clarify.

8. Table 4, why do the authors use notation P* instead of p* as in Table 3? What does P* mean?

9. Lines 173-175, the authors state “68.8% of isolates from C. tropicalis were high biofilm formers, showing statistically significant differences among Candida species according to its ability to form high biofilms (p < 0.0001).” So was C tropicalis statistically significantly highest compared to any other HBF species in Table 5? or was it just statistically significantly higher than a subset? Make clear what the subset is!

10. I do not understand what the authors are trying to say in lines 192-194, please re-write.

11. Lines 56-57, I am surprised that a sole blood culture can show biofilm infection? I would expect that for a biofilm related infection to be determined, then biofilm would have to be identified in an associated catheter or medical device or in tissue. This is similar to how a catheter-related-blood-stream-infection is diagnosed. Please describe how biofilm infection can be assessed solely from a blood sample.

12. Lines 213-214, the authors indicate a “lack of differentiation between planktonic and biofilm-related Candida infections in the diagnosis of the clinical laboratories at public health system” What is the current methodology used by these labs? Why were the authors able to differentiate between planktonic and biofilm infections?

Minor comments:

Abstract:

13. Lines 20 and 71, Please make some mention which databases were used for the literature review up front in the abstract and results, for example say “from publicly available data bases”. Right now, we do not learn which data bases were used until the methods section that comes at the end of the paper.

14. Line 23, Not sure the Q statistic ought to be reported in the abstract. If authors do want to leave it in, for the general PLOSONE audience, state what Q means and what it is used for.

15. Line 24 I^2 = 98.83% is huge! What does it mean?

16. Lines 26-27, two statements are made re: mortality (planktonic and biofilm), then the p-value < 0.0001 is stated. What test and which parameter does the p-value go with?

17. Line 29, what is meant by a low, medium or high biofilm? If you want to mention this in the abstract, then be clear here what is meant. The authors finally mention what is meant in the caption to Table 4 and line 163. Please explain how this determination “biofilm forming capacity” was made? Was it based on a categorization of the blood work outcome?

Intro:

18. Line 44, here the authors report on prevalence of nosocomial infections in the US. Since the meta-analysis focuses on Europe and Asia, please report Candida infection rate in Europe and Asia.

Results:

19. Line 75, Why and How were these 5 papers chosen? Figure 1 does not address these 5 or how they were chosen. This question is addressed later, but a brief explanation should be included here.

20. Table 2, I think by “All Candida spp. bloodstream infections” the authors mean planktonic associated infections, please clarify.

21. The sentence in lines 163-164 adds no more information than what is already in Table 4, I suggest removing this sentence.

Discussion:

22. Line 203, the statement re: “4/5” is not supported by Table 6. Table 6 instead suggests 70%. please clarify.

23. Line 307, the authors state “This meta-analysis is one of the few …” Where do the authors cite the other meta-analyses? Please refer to them here.

Methods:

24. Line 369, RStudio is a way to interact with R! It is OK to cite RStudio, but also cite the R software itself in the list of citations: R Core Team (2021). R: A language and environment for statistical computing. R Foundation for Statistical Computing, Vienna, Austria. URL https://www.R-project.org/.

25. Line 370, please properly cite the R packages. To find the citations, within R, use citation("meta"), etc.

26. Line 372, Please state what the random effect was. I am assuming the random effect was for study, but say so.

6. PLOS authors have the option to publish the peer review history of their article (what does this mean?). If published, this will include your full peer review and any attached files.

Reviewer #1: **Yes: **Elaine Cloutman-Green

Reviewer #2: No

---

## [Author Response · Author response to Decision Letter 0]

29 Nov 2021

Revised Manuscript

Answer to the Reviewer’s comments

Based on the following comments and suggestions, we have made new modifications to the previous original manuscript. These additional changes are also shown in the newly revised manuscript with track changes.

Comments to the Author

1. Is the manuscript technically sound, and do the data support the conclusions?

Reviewer #1: No

Reviewer #2: Yes

2. Has the statistical analysis been performed appropriately and rigorously? 

Reviewer #1: Yes

Reviewer #2: No

3. Have the authors made all data underlying the findings in their manuscript fully available?

Reviewer #1: Yes

Reviewer #2: No

4. Is the manuscript presented in an intelligible fashion and written in standard English?

Reviewer #1: Yes

Reviewer #2: Yes

5. Review Comments to the Author

Author’s answer – We are very grateful to both Reviewers for their constructive comments and thoughtful suggestions that really allowed us to improve the initial draft manuscript.

Reviewer 1 Report

Reviewer #1 (Doctor Elaine Cloutman-Green): 

Although this paper has merit in concept the authors appear to have tried to undertake a review that requires clinical knowledge without the background to do so. The ability of an organism to produce biofilm under laboratory conditions does not necessarily represent the condition of the organism under its clinically infectious state, especially when the review is aimed at only blood cultures, rather than taking into account infections linked to indwelling surfaces such as heart valves and prosthetics.

Author’s answer – We want to thank Doctor Elaine Cloutman-Green (Reviewer 1) for her constructive comments and thoughtful suggestions that really allowed us to improve the initial draft manuscript. Based on her comments and suggestions, we clarified confusing data and statements that initially we were not able to see in the original manuscript. We added a new table 2 in the revised manuscript describing the available information about the clinical background of the patients, which the authors evaluated the ability of Candida isolates to produce biofilm (Please check the new Table 2 on page 13 and the paragraph describing the available clinical background from the patients on page 15 lines 207-229 of the revised manuscript with track changes). It is well-known that the ability of a microorganism to produce biofilm in vitro does not necessarily represent the condition found in the infectious state of the patients; however, it is a virulence factor that may contribute to a more severe infection, and it is usually not detected in a standard evaluation of the isolated microorganism in the clinical procedures, which it was the main goal of the present study. Our results clearly correlated the ability to produce biofilm from Candida isolates with a higher mortality rate among patients with invasive candidiasis from different clinical backgrounds. Still, we recognized that the main drawbacks of the present study were the lack of a thorough clinical background from the patients and the diversity of the primary infections from the patients, such as infections linked to indwelling surfaces. Therefore, these limitations were added to the Results and Conclusions sections to better clarify the Readers (Please check these limitations on page 15 lines 207-209 at Results section and on page 28 lines 490-492 at Conclusions section of the revised manuscript with track changes). 

The authors attempt to compare levels of biofilm production between papers but do not comment on methodology utilised to assess biofilm production to support the reader in understanding whether they are reviewing comparable methods. Discussion of clinical presentations include items such as neonates and TPN without linking the fact that pre-term infants are likely to be on TPN and therefore this is correlation not causation. Is some of the data linked to considerable variation in mortality linked to this lack of clinical interpretation or patient factors or items such as drug availability. Were lines left in situ as a continuous source for instance - without information on whether the clinical analysis is comparable it is difficult to say, as noted with the laboratory comparisons, if the data is truly comparable.

Author’s answer – As well-appointed by Reviewer 1 (Doctor Elaine Cloutman-Green), although we stated in the Eligibility criteria section (on page 30 lines 536-537 of the revised manuscript with track changes) that all methods to quantify biofilm biomass was not a criterion to include or exclude any paper in this meta-analysis, we did not report the methodologies applied in these studies neither the biofilm classification criteria. Therefore, we added a new column in Table 1 with its respective legend and a new paragraph in the Overall effects of Candida biofilms section describing the methodologies applied in these studies and their biofilm classification criteria (Please check the new Table 1 on pages 7-8 and the new paragraph on pages 8-9 lines 142-150 of the revised manuscript with track changes). Although the methodologies to quantify biofilm biomass varied between studies, these methodologies are based on the optical density (OD) obtained by the combination of a certain colorimetric compound with the growth of the isolated Candida sp. and then it’s compared with reference strains in the same growth conditions. So, it is possible to compare results of biofilm production among Candida species between studies. The same procedure is also realized in biofilm classification criteria, but unfortunately, only 18 of the 31 studies realized this further evaluation, as reported in the Biofilm-forming capability in Candida spp. isolates section (Please check the description on page 18 lines 268-276 of the revised manuscript with track changes). Still, as described in the Discussion section, this diversity of methodologies could partially explain the inconsistency among the reports of Candida-associated biofilm infections in the published studies. Likewise, other numerous factors could also contribute to this heterogeneity of reports, such as the lack of differentiation between Candida species, the experience of the researchers, and the number of Candida isolates in the study set. These other factors were added in the Discussion section to avoid misunderstanding of the Readers (Please check the amendments on page 21 lines 339-344 of the revised manuscript with track changes). 

Concerning clinical presentations of the patients, we recognized that the original manuscript contained items and statements (such as neonates, TPN, and drug availability) lacking clinical interpretation or patient factors in the Results and Discussion sections. Several rectifications of these items and statements were realized in the revised manuscript trying to elucidate the gaps of the study and to avoid misunderstanding of the Readers (Please check these amendments on Results and Discussion sections on page 15 lines 207-222 and pages 22-24 lines 367-398, respectively, of the revised manuscript with track changes). However, as previously mentioned, these limitations were added to the Conclusions section to avoid misunderstanding of the Readers (Please check these limitations on page 28 lines 490-492 of the revised manuscript with track changes).

Therefore although the paper is of interest and could be modified for publication with further information I do not feel it is publishable in its current form.

Author’s answer – We are very grateful to Reviewer 1 for her constructive comments and thoughtful suggestions that really allowed us to improve the original manuscript. Several modifications were made to clarify the clinical background of the patients (the new Table 2, and the amendments in Results and Discussion sections) and the classification of biofilm production by Candida isolates in the selected studies (the new Table 1, and the amendments in Results section), also recognizing the limitations of the present meta-analysis. We hope that Reviewer 1 finds this version suitable for publication in PLOS ONE journal.

Reviewer 2 Report

Reviewer #2: 

The authors present a meta-analysis of Candida infections, and compare prevalence among geographic locations, Candida species, phenotypes (planktonic vs. biofilm; “biofilm forming capability”), and they also consider Candida resistance to a few anti-fungal agents.

There are some major issues, the most serious is that there are many confusing issues with the statistical diagnostics and reporting of statistical results. Secondly, the authors could provide a lot more details about how biofilm associated infections were diagnosed.

Author’s answer – We want to thank Reviewer 2 for the constructive comments and thoughtful suggestions that really allowed us to improve the original manuscript. As recommended by the Reviewer, several rectifications were made in the statistical diagnostics and results to clarify confusing issues and to avoid misunderstanding of the Readers. These amendments are detailed in the following answers to Reviewer 2 and illustrated in the revised manuscript with track changes.

1. Statements re: standard statistical tests for publication bias could be better worded. It’s great that the authors include the funnel plot so that readers can decide for themselves whether the plot suggests publication bias. Asymmetry IS evident in the plot, it is just that Egger’s test fails to find that the asymmetry is drastic enough to suggest publication bias. In light of this, the statement in lines 24-25 about the funnel plot and Egger's test showing no publication bias is incorrect (p=0.896), replace with a statement that says that Egger's test failed to show publication bias, or failed to find publication bias. Similarly, also change the incorrect wording on line 113, “the symmetry of the funnel plot confirmed the hypothesis of absence of bias,” it did not.

Author’s answer – As well-appointed by Reviewer 2, the standard tests for publication bias could be more accurately described and the presence of asymmetry is undeniable. To avoid misunderstanding by the Readers, we modified the original sentences on page 3 lines 43-44 and line 113 as suggested by Reviewer 2 (Please check these amendments on page 2 lines 30-31 and on page 11 lines 178-181 of the revised manuscript with track changes).

2. Even more worrisome is that the funnel plot does not show a funnel at all, what does this mean?!?! Are the authors conclusions still valid?

Author’s answer – As well-questioned by Reviewer 2, the funnel plot illustrated heterogeneity in the results among the studies of the data set. This heterogeneity does not invalidate the conclusions of the present manuscript. However, the high value of heterogeneity obtained in the pooled rate of biofilm formation in the data set, through the forest and funnel plots, needs to be carefully analyzed. 

As explained in the Discussion section of the revised manuscript with track changes, our Q and I2 values suggest high heterogeneity within each group, indicating that this heterogeneity is probably multifactorial, supported for a multiple meta-regression model results that explained more than 50% of the global variability as interacting variables. For example, if we consider only the articles from Italy, the sample size in 5 of 6 articles does not considerably differ but the effect size is quite different impacting, therefore, the funnel plot distribution. Furthermore, the asymmetry is probably not due to publication bias as the underlying cause, but to other information biases. Conclusions are still valid but should be interpreted with caution due to several factors that can be modulating them.

The triangular geometry of the funnel plot is basically a relationship between the effect-size and the error is associated with sampling distribution across studies. In our case, we found studies with low errors (similar sizes) but with drastic differences in the biofilm prevalence. This type of pattern probably indicates the presence of confounding variables (sub-groups undelaying structures), which are not included in the global analysis. Due to this high heterogeneity and considering the important observation of Reviewer 2, we include three segments in the Discussion section addressing this problem (Please check these explanations on pages 22-23 lines 372-385, on pages 24-25 lines 415-422, and on page 26 lines 455-460 of the revised manuscript with track changes) and we also included another segment describing a further meta-regression analysis using a combination of interacting variables in two models (Please check these results on pages 19-20 lines 304-308 of the revised manuscript with track changes).

References

Page, MJ, Sterne, JAC, Higgins, JPT, Egger, M. Investigating and dealing with publication bias and other reporting biases in meta-analyses of health research: A review. Res Syn Meth. 2021; 12: 248– 259. https://doi.org/10.1002/jrsm.1468

Sterne J A C, Sutton A J, Ioannidis J P A, Terrin N, Jones D R, Lau J et al. Recommendations for examining and interpreting funnel plot asymmetry in meta-analyses of randomised controlled trials BMJ 2011; 343: d4002.

https://doi.org/10.1136/bmj.d4002

3. For the PLOS ONE audience, please explain why the funnel plot should look like a funnel and explain the axes.

Author’s answer – As well-recommended by Reviewer 2, the brief and simple explanation about the funnel plot, their axes, and expected results were written for the Readers of PLOS ONE. More exactly, we add a summary of the funnel plot on page 12 lines 185-192 of the revised manuscript with track changes with the following text: 

“The funnel plot of this study illustrates the effect size (biofilm prevalence) on the x-axis and the standard error (SE) on the y-axis. In case of no publication bias in the data set, the studies are distributed evenly around the pooled effect size. The smaller studies should appear near the bottom due to their higher variance when compared to the larger studies, which should be placed at the top of the plot. The diagonal lines show the expected 95% confidence intervals around the summary estimate. In the absence of heterogeneity, the studies of the data set should lie within the funnel defined by these diagonal lines. However, heterogeneity and some asymmetries among the studies of the data set were illustrated by the funnel plot.”

Finally, an explanation of each axis was also added below Fig. 3 (Please check this description on page 11 lines 176-177 of the revised manuscript with track changes) with the following short text: 

“Studies are represented by a point. The X-axis represents the effect size (biofilm prevalence), and the Y-axis shows the standard error.”

Reference 

Sterne JA, Egger M. Funnel plots for detecting bias in meta-analysis: guidelines on choice of axis. J Clin Epidemiol. 2001 Oct;54(10):1046-55. doi: 10.1016/s0895-4356(01)00377-8. PMID: 11576817.

4. Line 100, what does the p-value < 0.001 associate with? Not the t-stat of 0.387! The Q stat is the test statistic for the test of heterogeneity across studies. What is the t-stat for? Both these stats have associated p-values, but only one is reported here.

Author’s answer – As questioned by Reviewer 2, the p-value < 0.001 referred to the prevalence of biofilm rate in the data set, showing a statistical prevalence. This explanation was clarified in the original sentence, and we also replaced the t-stat value with the �2 value (�2 = 0.150). The initial p-value mentioned is associated with the model, which justifies the random-effect analysis. The t-test was not used, but rather tau-squared which estimates the variance between the effect sizes of the studies in the model. Unfortunately, there is still no consensus on which cut-off points to use to interpret this test. Finally, we also included the p-value of Cochran's Q (Please check these amendments on page 9 lines 157-161 of the revised manuscript with track changes).

5. Tables 2 and 3, What are k, P, p*, Q, I and $\\tau$ in the table? Is k the number of studies? Is P the p-value for Egger's test? For the general PLOS audience not familiar with meta-analyses, please provide brief summaries of these.

Author’s answer – As well-proposed by Reviewer 2, a summary of each statistic symbol was included in the section “Data analysis and statistical methods” and the meaning of each statistic symbol was added in the legends below the tables for the general PLOS audience not familiar with meta-analyses (Please check these legends now on tables 3 and 4 at pages 16-17 and the brief summary on page 31 lines 561-569 of the revised manuscript with track changes). Lastly, the k is the number of studies, the p-value (in lower case) mentioned corresponds to Egger's test, and the p*-value corresponds to the random-effect model significance level. It is important to mention that the notation of P* was wrongly put in the tables due to automated correction of p* (in lower case) and we apologize for this mistake.

6. Lines 149-150, “Although the biofilm prevalence varied among regions, no statistically significant value was obtained in this subgroup analysis” is uninformative. Consider replacing with “Although the biofilm prevalence varied among regions, there were no statistically significant differences (p = 0.4049).”

Author’s answer – As advised by Reviewer 2, the sentence from lines 149-150 was replaced by “Although the biofilm prevalence varied among regions, there were no statistically significant differences (p = 0.4049).” (Please check this adjustment on page 17 lines 257-258 of the revised manuscript with track changes). 

7. Lines 151-152, the authors are keen to point out getting a small p-value, “Meanwhile, when comparing prevalence rates between countries, a statistically significant value was obtained (p = 0.0074)”. The authors should make clear WHY there was small p-value. That p-value says there is some statistically significant difference between at least 2 countries, but the authors do not make clear which countries are statistically significantly different. In lines 153 and 154 the authors mention that Italy has the lowest rate and that Spain has the highest. Is Spain statistically significantly the highest than all other countries? Or maybe Spain is statistically significantly higher than just Italy? The authors make a nebulous reference to this comparison again in lines 236-237 which seems to suggest that Spain is statistically significantly higher than just Italy. Please clarify.

Author’s answer – As well-appointed by Reviewer 2, pairwise comparisons were realized between Spain and other countries demonstrating that Spain was statistically superior to Brazil (p <0 .0001), Italy (p = 0.0263), and India (p = 0.0030). We clarified these results on page 17 lines 261-264 in the Results section and also on page 24 lines 401-404 in the Discussion section. The p-value corresponds to the test for pairwise compared subgroup differences.

8. Table 4, why do the authors use notation P* instead of p* as in Table 3? What does P* mean?

Author’s answer – As previously answered to Reviewer 2, the notation of P* was wrongly put in the tables due to the automated correction of p* (in lower case). We rectified this error, and we apologize for this mistake (Please check this rectification now on Table 5 on page 18 of the revised manuscript with track changes).

9. Lines 173-175, the authors state “68.8% of isolates from C. tropicalis were high biofilm formers, showing statistically significant differences among Candida species according to its ability to form high biofilms (p < 0.0001).” So was C tropicalis statistically significantly highest compared to any other HBF species in Table 5? or was it just statistically significantly higher than a subset? Make clear what the subset is!

Author’s answer – As well-observed by Reviewer 2, C. tropicalis was statistically higher in this subgroup analysis between all Candida species. To better clarify the Readers, we recalculated and performed a pairwise comparison between each of the Candida species clarifying the original sentence (Please check this rectification now on page 19 lines 294-296 of the revised manuscript with track changes), more exactly:

 “More precisely, the HBF prevalence of C. tropicalis was the highest showing statistically significant differences with the other Candida species, except for C. krusei (p = 0.5477) and C. glabrata (p = 0.0896)”.

10. I do not understand what the authors are trying to say in lines 192-194, please re-write.

Author’s answer – We apologized for this confusing sentence. As well-suggested by Reviewer 2, we re-wrote the original sentence to avoid misunderstanding by the Readers (Please check the new sentence on page 20 lines 321-325 of the revised manuscript with track changes), more exactly: 

“When comparing to planktonic cells, Candida-related biofilm isolates showed a statistical increment of resistance against the three antifungals evaluated in the study (p < 0.001).”

11. Lines 56-57, I am surprised that a sole blood culture can show biofilm infection? I would expect that for a biofilm related infection to be determined, then biofilm would have to be identified in an associated catheter or medical device or in tissue. This is similar to how a catheter-related-blood-stream-infection is diagnosed. Please describe how biofilm infection can be assessed solely from a blood sample.

Author’s answer – As wished by Reviewer 2, a more detailed explanation was added about how biofilm infection was assessed from a blood sample (Please check this description on pagse 4-5 lines 88-93 of the revised manuscript with track changes). Briefly, the selected studies performed an in vitro biofilm assay using Candida isolates from blood samples of patients. It is important to mention that the study set of the present meta-analysis included patients with catheter-related candidemia (CRC) and non-CRC. In cases of patients with CRC, the standard procedure was blood cultures from obtained the catheter and peripheral veins, whereas non-CRC was indicated by the recovery of Candida spp. from only blood samples, as previously described by Guembe et al. (2014). Finally, a better portrayal of the methodologies applied in the selected studies was added to Table 1 and in the Results section to better clarify for the Readers (Please check the new Table 1 on pages 7-8 and the new paragraph on pages 8-9 lines 142-150 of the revised manuscript with track changes). 

Reference

Guembe M, Guinea J, Marcos-Zambrano L, Fernández-Cruz A, Peláez T, Muñoz P, Bouza E. Is biofilm production a predictor of catheter-related candidemia? Med Mycol. 2014 May;52(4):407-10. doi: 10.1093/mmy/myt031. Epub 2014 Apr 28. PMID: 24782103.

12. Lines 213-214, the authors indicate a “lack of differentiation between planktonic and biofilm-related Candida infections in the diagnosis of the clinical laboratories at public health system” What is the current methodology used by these labs? Why were the authors able to differentiate between planktonic and biofilm infections?

Author’s answer – As well-questioned by Reviewer 2 and previously discussed by Høiby and colleagues (2015), the traditional clinical microbiology laboratories have focused on culturing and testing planktonically (=non-aggregated) growing microorganisms and have reported the susceptibility to various antibiotics and antiseptics under planktonic growth conditions. 

Meanwhile, the authors of selected studies of this manuscript applied a further analysis by evaluating the ability of biofilm production in Candida isolates through an in vitro biofilm assay. Furthermore, in Candida biofilms, traditional techniques require device removal followed by culture or microscopy of a catheter segment, while catheter-sparing diagnostic tests include paired quantitative blood cultures. However, as previously indicated by Høiby et al. (2015) and Bouza et al. (2013), the number of positive peripheral blood cultures also seems to be a promising diagnostic tool to diagnose catheter-related candidemia without directly removing the catheter.

Finally, all these clarifications were added in the revised manuscript (Please check this description on pages 21-22 lines 346-355 of the revised manuscript with track changes).

References

Høiby N, Bjarnsholt T, Moser C, Bassi GL, Coenye T, Donelli G, Hall-Stoodley L, Holá V, Imbert C, Kirketerp-Møller K, Lebeaux D, Oliver A, Ullmann AJ, Williams C; ESCMID Study Group for Biofilms and Consulting External Expert Werner Zimmerli. ESCMID guideline for the diagnosis and treatment of biofilm infections 2014. Clin Microbiol Infect. 2015 May;21 Suppl 1:S1-25. doi: 10.1016/j.cmi.2014.10.024. Epub 2015 Jan 14. PMID: 25596784.

Bouza E, Alcalá L, Muñoz P, Martín-Rabadán P, Guembe M, Rodríguez-Créixems M; GEIDI and the COMIC study groups. Can microbiologists help to assess catheter involvement in candidaemic patients before removal? Clin Microbiol Infect. 2013 Feb;19(2):E129-35. doi: 10.1111/1469-0691.12096. Epub 2012 Dec 10. PMID: 23231412.

Minor comments:

Abstract:

13. Lines 20 and 71, Please make some mention which databases were used for the literature review up front in the abstract and results, for example say “from publicly available data bases”. Right now, we do not learn which data bases were used until the methods section that comes at the end of the paper.

Author’s answer– As well-recommended by Reviewer 2, we mentioned the databases used in the study, more exactly:

Original line 20 (now on page 2 lines 37-38 of the revised manuscript with track changes): “A total of 31 studies from publicly available databases met our inclusion criteria.”

Original line 71 (now on page 5 lines 108-109 of the revised manuscript with track changes): “A total of 214 studies were retrieved and 70 full texts were reviewed from publicly available databases (Web of Science, Scopus, PubMed, and Google Scholar).”

Finally, it is important to mention that the Abstract section was rewritten following the PLOS ONE guidelines for a meta-analysis study (Please check the new Abstract section on pages 2-3 of the revised manuscript with track changes).

14. Line 23, Not sure the Q statistic ought to be reported in the abstract. If authors do want to leave it in, for the general PLOSONE audience, state what Q means and what it is used for.

Author’s answer– We decided to maintain the Q statistic because it is standard procedure in most meta-analysis studies that we read before. However, as suggested by Reviewer 2, we added the indication that Q is related to the heterogeneity of the results together with I2 and t2, indicating the high heterogeneity obtained through the random-effects model. 

Original line 23 (now on page 2 lines 39-41 of the revised manuscript with track changes): “Forest plot showed a pooled rate of biofilm formation of 80.0 % (CI: 67–90), with high heterogeneity (Q = 2567.45, I2 = 98.83, t2 = 0.150) in random effects model (p<0.001).”

15. Line 24 I^2 = 98.83% is huge! What does it mean?

Author’s answer– As already mentioned in the responses to comments number 2 and 3 of Reviewer 2, the high heterogeneity is probably due to multifactorial causes, which is supported by the results of our meta-regression model. As clarified on pages 19-20 lines 304-308 of the revised manuscript with track changes, this meta-regression model explains more than 50% of the overall variability as interacting variables, specifically geographical distribution, and Candida species. In addition, it is also reported that the lack of information about the clinical background among patients in the selected studies (data set) is the main drawback and may be an important source of unexplained variation in the present meta-analysis. These results are extensively explained in several paragraphs in the Discussion section, more exactly, on pages 22-23 lines 372-385, on pages 24-25 lines 415-422, and on page 26 lines 455-460 of the revised manuscript with track changes.

16. Lines 26-27, two statements are made re: mortality (planktonic and biofilm), then the p-value < 0.0001 is stated. What test and which parameter does the p-value go with?

Author’s answer– As well-appointed by Reviewer 2, the p-value < 0.0001 was stated in the Abstract section and leading to misunderstanding by the Readers. The p-value < 0.0001 was obtained through the random effect model significance level, which is already clarified in Table 2 (now Table 3 in the revised manuscript) and associated with the mortality rate in biofilm-associated infections. However, we re-wrote the original sentence to avoid misinterpretation by the Readers (Please check this description on page 3 lines 44-45 of the revised manuscript with track changes), more exactly:

“The mortality rate in Candida-related bloodstream infections was 37.9% of which 70.0% were from biofilm-associated infections.”

17. Line 29, what is meant by a low, medium or high biofilm? If you want to mention this in the abstract, then be clear here what is meant. The authors finally mention what is meant in the caption to Table 4 and line 163. Please explain how this determination “biofilm forming capacity” was made? Was it based on a categorization of the blood work outcome?

Author’s answer– As suggested by Reviewer 2, we clarified that the classification of Candida isolates in low, intermediate, or high biofilm formers was based on the level of biofilm mass (crystal violet staining or XTT assays) in the Abstract section (Please check this description on page 3 lines 46-48 of the revised manuscript with track changes). Also, we explained how the classification of Candida isolates was made in the studies of the data set in the Results section (Please check this explanation on page 18 lines 268-275 of the revised manuscript with track changes), more exactly: 

“Briefly, biofilm forming capacity was assessed using the crystal violet or XTT assays, measuring the biofilm mass. Candida isolates were cultured in 96-well plates at 37°C for 24 h and the biomass of each isolate was measured. Then, isolates were grouped based on their level of biomass, more exactly: low biofilm formers (LBF) showed a biomass production below the 1st quartile (Q1; Absisolate < 0.432), intermediate biofilm formers (IBF) evidenced a biomass production in the 2nd quartile (Q2; 0.432 < Absisolate < 1.07), and high biofilm formers (HBF) demonstrated a biomass production higher the 1st quartile 3rd quartile (Q3; Absisolate > 1.07), as previously described by Monfredini et al. (2018) and Vitális et al. (2020).”

References

Monfredini PM, Souza ACR, Cavalheiro RP, Siqueira RA, Colombo AL. Clinical impact of Candida spp. biofilm production in a cohort of patients with candidemia. Med Mycol. 2018 Oct 1;56(7):803-808. doi: 10.1093/mmy/myx133. PMID: 29228246.

Vitális E, Nagy F, Tóth Z, Forgács L, Bozó A, Kardos G, Majoros L, Kovács R. Candida biofilm production is associated with higher mortality in patients with candidaemia. Mycoses. 2020 Apr;63(4):352-360. doi: 10.1111/myc.13049. Epub 2020 Jan 23. PMID: 31943428.

Intro:

18. Line 44, here the authors report on prevalence of nosocomial infections in the US. Since the meta-analysis focuses on Europe and Asia, please report Candida infection rate in Europe and Asia.

Author’s answer– As suggested by Reviewer 2, we added the report on candidemia incidence in Europe and Asia, as well as one study realized in Latin America evolving seven countries (Please check this information on page 4 lines 70-76 of the revised manuscript with track changes), more exactly:

“In Europe, Bassetti and colleagues realized a multinational and multicenter study in 2019 reporting 7.07 episodes per 1000 in European intensive care units (ICUs) with a 30-day mortality of 42% [3]. While, in the Asia-Pacific region, Hsueh and colleagues reported a candidemia incidence in ICUs of 5- to 10-fold higher than in the entire hospital and a mortality rate of patients between 35% and 60% [4]. In Latin America, Nucci and colleagues realized a laboratory-Based Survey between November 2008 and October 2010 among 20 tertiary care hospitals in seven Latin American countries, reporting an overall incidence of 1.18 cases per 1,000 in general admissions [5].”

References

Bassetti, M., Giacobbe, D.R., Vena, A. et al. Incidence and outcome of invasive candidiasis in intensive care units (ICUs) in Europe: results of the EUCANDICU project. Crit Care 23, 219 (2019). https://doi.org/10.1186/s13054-019-2497-3.

Hsueh PR, Graybill JR, Playford EG, Watcharananan SP, Oh MD, Ja'alam K, Huang S, Nangia V, Kurup A, Padiglione AA. Consensus statement on the management of invasive candidiasis in Intensive Care Units in the Asia-Pacific Region. Int J Antimicrob Agents. 2009 Sep;34(3):205-9. doi: 10.1016/j.ijantimicag.2009.03.014. Epub 2009 May 5. PMID: 19409759.

Nucci M, Queiroz-Telles F, Alvarado-Matute T, Tiraboschi IN, Cortes J, Zurita J, et al. Epidemiology of Candidemia in Latin America: A Laboratory-Based Survey. PLoS One. 2013;8: e59373. doi:10.1371/journal.pone.0059373

Results:

19. Line 75, Why and How were these 5 papers chosen? Figure 1 does not address these 5 or how they were chosen. This question is addressed later, but a brief explanation should be included here.

Author’s answer– As well-recommended by Reviewer 2, we added a brief explanation about the selection of at least 5 or more papers to realize subgroup analysis using a random-effect model and to answer other relevant questions about Candida-related biofilms (such as the mortality rate related to biofilms, the geographical distribution of biofilms, the characterization of biofilm production among Candida species, and the correlation between biofilm formation and antifungal resistance). Please check this explanation on page 6 lines 115-119 of the revised manuscript with track changes.

20. Table 2, I think by “All Candida spp. bloodstream infections” the authors mean planktonic associated infections, please clarify.

Author’s answer– As suggested by Reviewer 2, we clarified the information about “All Candida spp. bloodstream infections” in the text before the original Table 2 (now Table 3 in the revised manuscript), more exactly:

Original lines 129-131 (now on page 16 lines 234-237 of the revised manuscript with track changes) 

“As shown in Table 3, the pooled mortality rate due to Candida-related bloodstream infections was 37.9% (95% CI: 26.2-50.2) of which the mortality associated with biofilm-forming infections was 70.0% (95% CI: 52.8–84.8).”

21. The sentence in lines 163-164 adds no more information than what is already in Table 4, I suggest removing this sentence.

Author’s answer– As advised by Reviewer 2, we removed the sentence on lines 163-164 of the original manuscript (Please check this deletion on page 18 lines 281-283 of the revised manuscript with track changes).

Discussion:

22. Line 203, the statement re: “4/5” is not supported by Table 6. Table 6 instead suggests 70%. please clarify.

Author’s answer– As well-observed by Reviewer 2, we apologized for this mistake. It is around 70.0% and not 80.0% as stated in the sentence of the Discussion section. We rectified the sentence in the original line 203 to avoid misunderstanding by the Readers (Please check the new sentence on page 21 lines 334-335 of the revised manuscript with track changes).

23. Line 307, the authors state “This meta-analysis is one of the few …” Where do the authors cite the other meta-analyses? Please refer to them here.

Author’s answer– As suggested by Reviewer 2, we cited the other meta-analyses in the sentence of the Conclusions section (Please check these references in the sentence on page 28 lines 494-497 of the revised manuscript with track changes), more exactly:

“To the authors’ best knowledge, this meta-analysis is one of the few that explored the association of biofilm production among different Candida species in bloodstream infections [64–67], using data published worldwide and adhering to the Preferred Reporting Items for Systematic Reviews and Meta-Analyses guideline.”

References

64. Pammi M, Holland L, Butler G, Gacser A, Bliss JM. Candida parapsilosis is a significant neonatal pathogen: a systematic review and meta-analysis. Pediatr Infect Dis J. 2013 May;32(5):e206-16. doi: 10.1097/INF.0b013e3182863a1c. PMID: 23340551; PMCID: PMC3681839.

65. Buehler SS, Madison B, Snyder SR, Derzon JH, Cornish NE, Saubolle MA, Weissfeld AS, Weinstein MP, Liebow EB, Wolk DM. Effectiveness of Practices To Increase Timeliness of Providing Targeted Therapy for Inpatients with Bloodstream Infections: a Laboratory Medicine Best Practices Systematic Review and Meta-analysis. Clin Microbiol Rev. 2016 Jan;29(1):59-103. doi: 10.1128/CMR.00053-14. PMID: 26598385; PMCID: PMC4771213.

66. Kobayashi T, Marra AR, Schweizer ML, Ten Eyck P, Wu C, Alzunitan M, Salinas JL, Siegel M, Farmakiotis D, Auwaerter PG, Healy HS, Diekema DJ. Impact of Infectious Disease Consultation in Patients With Candidemia: A Retrospective Study, Systematic Literature Review, and Meta-analysis. Open Forum Infect Dis. 2020 Aug 3;7(9):ofaa270. doi: 10.1093/ofid/ofaa270. PMID: 32904995; PMCID: PMC7462368.

67. Pinto H, Simões M, Borges A. Prevalence and Impact of Biofilms on Bloodstream and Urinary Tract Infections: A Systematic Review and Meta-Analysis. Antibiotics (Basel). 2021 Jul 8;10(7):825. doi: 10.3390/antibiotics10070825. PMID: 34356749; PMCID: PMC8300799. 

Methods:

24. Line 369, RStudio is a way to interact with R! It is OK to cite RStudio, but also cite the R software itself in the list of citations: R Core Team (2021). R: A language and environment for statistical computing. R Foundation for Statistical Computing, Vienna, Austria. URL https://www.R-project.org/.

Author’s answer– As well-appointed by Reviewer 2, RStudio is only a way to interact with R. So we rectified the sentence also citing R software (Please check the amendment and this reference in the sentence on page 31 lines 559-561 of the revised manuscript with track changes), more exactly:

“Meta-analysis was performed using several R packages ("meta", "metafor", "poibin", and "stringr") of R version 3.4.3 [70] and RStudio version 1.3.1073 [71] (S3 File) .”

References

70. R Core Team. R: A language and environment for statistical computing. Vienna (Austria): R Foundation for Statistical Computing; 2021. Available: https://www.r-project.org/.

71. RStudio Team. RStudio: Integrated Development for R. Boston (USA): RStudio, Inc.; 2021. Available: http://www.rstudio.com/

25. Line 370, please properly cite the R packages. To find the citations, within R, use citation("meta"), etc.

Author’s answer– As previously answered to Reviewer 2, the original sentence was rectified properly citing R software and the R packages (Please check the amendment and this reference in the sentence on page 31 lines 559-561 of the revised manuscript with track changes), more exactly:

“Meta-analysis was performed using several R packages ("meta", "metafor", "poibin", and "stringr") of R version 3.4.3 [70] and RStudio version 1.3.1073 [71] (S3 File).”

26. Line 372, Please state what the random effect was. I am assuming the random effect was for study, but say so.

Author’s answer– As advised by Reviewer 2, we rectified the sentence by stating that the random-effects model was used for the study (Please check the amendment on page 31 lines 566-568 of the revised manuscript with track changes).

Author’s answer – We appreciated all suggestions and efforts made by Reviewer 2 that really allowed us to improve the initial draft manuscript. We hope that Reviewer 2 finds this version suitable for publication in PLOS ONE journal.

---

## [Decision Letter · Decision Letter 1]

10 Jan 2022

PONE-D-21-23498R1Prevalence of biofilms in Candida spp. bloodstream infections: a meta-analysisPLOS ONE

Dear Dr. Antonio Machado,

Thank you for submitting your manuscript to PLOS ONE. After careful consideration, we feel that it has merit but does not fully meet PLOS ONE’s publication criteria as it currently stands. Therefore, we invite you to submit a revised version of the manuscript that addresses the points raised during the review process.

We look forward to receiving your revised manuscript.

Kind regards,

Surasak Saokaew, PharmD, PhD, BPHCP, FACP, FCPA

Academic Editor

PLOS ONE

Reviewers' comments:

Reviewer's Responses to Questions

**Comments to the Author**

1. If the authors have adequately addressed your comments raised in a previous round of review and you feel that this manuscript is now acceptable for publication, you may indicate that here to bypass the “Comments to the Author” section, enter your conflict of interest statement in the “Confidential to Editor” section, and submit your "Accept" recommendation.

Reviewer #2: (No Response)

2. Is the manuscript technically sound, and do the data support the conclusions?

Reviewer #2: Yes

3. Has the statistical analysis been performed appropriately and rigorously? 

Reviewer #2: Yes

4. Have the authors made all data underlying the findings in their manuscript fully available?

Reviewer #2: Yes

5. Is the manuscript presented in an intelligible fashion and written in standard English?

Reviewer #2: Yes

6. Review Comments to the Author

Reviewer #2: The authors did a very good job addressing the issues from the first review. The only remaining suggestion I have is to properly cite the R packages in the methods section, and add the following references to the bibliography:

For "meta"

Balduzzi S, Rücker G, Schwarzer G (2019). “How to perform a meta-analysis with R: a practical tutorial.” Evidence-Based Mental Health, 153–160.

For "metafor"

Viechtbauer W (2010). “Conducting meta-analyses in R with the metafor package.” Journal of Statistical Software, 36(3), 1–48. https://doi.org/10.18637/jss.v036.i03.

For "poibin"

Hong, Y. (2013). On computing the distribution function for the Poisson binomial distribution. Computational Statistics & Data Analysis, Vol. 59, pp. 41-51.

For "stringr"

Wickham H (2021). stringr: Simple, Consistent Wrappers for Common String Operations. http://stringr.tidyverse.org, https://github.com/tidyverse/stringr.

7. PLOS authors have the option to publish the peer review history of their article (what does this mean?). If published, this will include your full peer review and any attached files.

Reviewer #2: No

---

## [Author Response · Author response to Decision Letter 1]

10 Jan 2022

Reviewer 2 Report

Author’s answer –We thank Reviewer 2 for their insight and supportive comments. These amendments are detailed in the following answer to Reviewer 2 and illustrated in the revised manuscript with track changes. We hope that Reviewer 2 finds this version suitable for publication in PLOS ONE journal.

Reviewer #2: 

6. Review Comments to the Author

Reviewer #2: The authors did a very good job addressing the issues from the first review. The only remaining suggestion I have is to properly cite the R packages in the methods section, and add the following references to the bibliography:

For "meta"

Balduzzi S, Rücker G, Schwarzer G (2019). “How to perform a meta-analysis with R: a practical tutorial.” Evidence-Based Mental Health, 153–160.

For "metafor"

Viechtbauer W (2010). “Conducting meta-analyses in R with the metafor package.” Journal of Statistical Software, 36(3), 1–48. https://doi.org/10.18637/jss.v036.i03.

For "poibin"

Hong, Y. (2013). On computing the distribution function for the Poisson binomial distribution. Computational Statistics & Data Analysis, Vol. 59, pp. 41-51.

For "stringr"

Wickham H (2021). stringr: Simple, Consistent Wrappers for Common String Operations. http://stringr.tidyverse.org, https://github.com/tidyverse/stringr.

Author’s answer– As well-appointed by Reviewer 2, we properly cited the R packages in the methods section and added the references to the bibliography (Please check these amendments on page 30 lines 518-519 and page 41 lines 745-753 of the revised manuscript with track changes, respectively). Also, we want to thank Reviewer 2 to give us the proper citations of the R packages that allowed us to quickly rectify our mistake.

---

## [Editor Report · Decision Letter 2]

21 Jan 2022

Prevalence of biofilms in Candida spp. bloodstream infections: a meta-analysis

PONE-D-21-23498R2

Dear Dr. Antonio Machado,

We’re pleased to inform you that your manuscript has been judged scientifically suitable for publication and will be formally accepted for publication once it meets all outstanding technical requirements.

Kind regards,

Surasak Saokaew, PharmD, PhD, BPHCP, FACP, FCPA

Academic Editor

PLOS ONE
---

## [Editor Report · Acceptance letter]

25 Jan 2022

PONE-D-21-23498R2 

Prevalence of biofilms in *Candida* spp. bloodstream infections: a meta-analysis 

Dear Dr. Machado:

I'm pleased to inform you that your manuscript has been deemed suitable for publication in PLOS ONE. Congratulations! Your manuscript is now with our production department. 

Kind regards, 

on behalf of

Dr. Surasak Saokaew 

Academic Editor

PLOS ONE